# Ether phospholipids are required for mitochondrial reactive oxygen species homeostasis

Ziheng Chen [1] ✉, I-Lin Ho[1], Melinda Soeung [1], Er-Yen Yen [1], Jintan Liu [1], Liang Yan[2], Johnathon L. Rose[1,3], Sanjana Srinivasan[1,3], Shan Jiang[3], Q. Edward Chang[4], Ningping Feng[5], Jason P. Gay[5], Qi Wang [6], Jing Wang [6], Philip L. Lorenzi [6], Lucas J. Veillon[6], Bo Wei[6], John N. Weinstein [6,7], Angela K. Deem[3], Sisi Gao[1], Giannicola Genovese[1,8], Andrea Viale[1], Wantong Yao[9], Costas A. Lyssiotis [10,11,12], Joseph R. Marszalek [3,5], Giulio F. Draetta [1,2] ✉ & Haoqiang Ying [2] ✉

Mitochondria are hubs where bioenergetics, redox homeostasis, and anabolic metabolism pathways integrate through a tightly coordinated flux of metabolites. The contributions of mitochondrial metabolism to tumor growth and therapy resistance are evident, but drugs targeting mitochondrial metabolism have repeatedly failed in the clinic. Our study in pancreatic ductal adenocarcinoma (PDAC) finds that cellular and mitochondrial lipid composition influence cancer cell sensitivity to pharmacological inhibition of electron transport chain complex I. Profiling of patient-derived PDAC models revealed that monounsaturated fatty acids (MUFAs) and MUFA-linked ether phospholipids play a critical role in maintaining ROS homeostasis. We show that ether phospholipids support mitochondrial supercomplex assembly and ROS production; accordingly, blocking de novo ether phospholipid biosynthesis sensitized PDAC cells to complex I inhibition by inducing mitochondrial ROS and lipid peroxidation. These data identify ether phospholipids as a regulator of mitochondrial redox control that contributes to the sensitivity of PDAC cells to complex I inhibition.

Metabolism reprogramming is recognized as one of the hallmarks of human cancers[1,2]. In general, tumor cell metabolism features activate biosynthetic pathways to support oncogenic growth. Tumor cells also strengthen defense mechanisms against various environmental or endogenous insults, such as oxidative stress, which can be a consequence of activated metabolism and growth programs. However, extensive crosstalk between metabolic pathways ensures that cellular metabolism is highly plastic and adaptive, making tumor metabolism challenging to target therapeutically.

Mitochondria are hubs where metabolic pathways converge. Here, bioenergetics, anabolic metabolism, and redox homeostasis are integrated by the tightly coordinated flux of metabolites through the tricarboxylic acid (TCA) cycle and through the oxidative phosphorylation (OXPHOS) pathway mediated by the electron transport chain (ETC). Although hyperactivation of glycolysis (the Warburg effect) was historically viewed as the most relevant metabolic adaptation in tumor cells, the essential role of mitochondrial metabolism to fuel tumor growth is now evident in multiple contexts. It has been established in multiple cancer types that mitochondrial OXPHOS drives resistance to chemotherapy and a variety of targeted therapies[3,4]. In pancreatic ductal adenocarcinoma (PDAC), a subset of cells with stem cell-like properties relies on OXPHOS[5]. Various strategies to target

mitochondrial metabolism are in pre-clinical or clinical development;[6] however, therapeutic targeting of OXPHOS in the clinical setting has failed due to either toxicity or lack of therapeutic response. Given the promise of metabolism-targeting drugs to treat cancer, there is a clear need to delineate the dependency of tumor cells on mitochondrial function as well as the adaptive mechanisms that enable escape from OXPHOS inhibition.

In this study, we identify a unique differential dependency on mitochondrial complex I activity among human PDAC cells and find that upregulation of peroxisome-derived ether phospholipids, especially those conjugated with monounsaturated fatty acids (MUFAs), enable PDAC cells to adapt to and survive mitochondrial complex I inhibition. We further demonstrate a direct relationship between the regulation of mitochondrial ROS and lipid peroxidation that affects tumor cell viability. Our findings provide a mechanistic rationale for strategies that target multiple axes of mitochondrial metabolism to effectively kill PDAC cells.

## Results

### Complex I inhibition identifies a latent vulnerability in a subset of PDAC models

To investigate the dependence on mitochondrial OXPHOS in PDAC, we evaluated the sensitivity of human PDAC cells derived from patient-derived xenograft (PDX) models, which faithfully recapitulate the histology and molecular heterogeneity of human PDAC[7–9]. We evaluated OXPHOS inhibitors targeting different mitochondrial complexes, including the complex I inhibitors metformin[10], phenformin[11], and IACS-010759;[12,13] the complex III inhibitor antimycin A[14] (targeting cytochrome c reductase); and the complex V ATP synthase inhibitor, oligomycin A[15] (Fig. 1a–c, Supplementary Fig. 1a–e). Each OXPHOS inhibitor was evaluated in multiple cell lines derived from PDX models (PDX lines). We detected differential sensitivity to complex I inhibitors among the models when cultured in limited-glucose (5.5 mM) medium that was not observed for complex III or complex V inhibitors (Fig. 1a-c, Supplementary Fig. 1a–e). Further testing of the effects of IACS-010759 showed that, in glucose-rich medium (25 mM), all PDX lines maintained viability when exposed to IACS-010759, whereas significant cell death was observed in PDX lines cultured in limited-glucose medium (Supplementary Fig. 2a, Supplementary Table 1), supporting the role of glycolysis as a compensatory mechanism for OXPHOS inhibition[16]. Oxygen consumption following exposure to IACS-010759 in limited-glucose medium was similarly attenuated in all four PDX lines (Fig. 1d–g), suggesting that a unique program enables survival even with severely impaired ETC complex I function and in a glucose-depleted environment. Interestingly, complex I inhibition-resistant PDX lines (hereafter, resistant PDX lines) were able to maintain viability even upon complete depletion of glucose from the culture medium (Supplementary Fig. 2a, b). Further, the differential sensitivity to complex I inhibition among the PDX lines was maintained in galactose culture to inhibit glycolysis dependency as well as when cultured in hypoxic conditions (Supplementary Fig. 3a, b), indicating that sensitivity to complex I inhibition is affected by glycolysis activity, but is not dictated by oxygen level. Notably, we did not observe any correlation between the most frequently occurring genetic lesions in PDAC, including mutation/deletion of *KRAS*, *TP53*, *CDKN2A/P16* and *SMAD4*, and sensitivity to complex I inhibition (Supplementary Table 2).

To validate responses to complex I inhibition in vivo, we selected IACS-010759 as an optimal complex I inhibitor owing to its potency and selectivity as well as superior in vivo pharmacology[17], and we evaluated its impact on tumor growth in nude mice inoculated with different PDX lines. Upon tumor establishment, animals began a fasting-feeding protocol that has been shown to enhance metformin-induced tumor suppression (Supplementary Fig. 4a)[18], and they were randomized to receive treatment with IACS-010759 (5 mg/kg/every other day) or vehicle (*n* = 4 to 9 animals per group). In all PDX models, treatment with IACS-010759 reduced tissue hypoxia, as measured by pimonidazole (Supplementary Fig. 4b), indicating that the impaired cellular oxygen consumption allowed oxygen accumulation in all tumors, independent of any effect on growth. As anticipated, fasting induced a decrease in blood glucose concentration in both treatment groups (Supplementary Fig. 4c). The in vivo response of tumors derived from PDX lines mirrored the in vitro findings: IACS-010759 treatment significantly suppressed growth of tumors derived from PDX lines that were sensitive to IACS-010759 in vitro but had no effect on tumors derived from IACS-010759-resistant PDX lines (Fig. 1h–k). Accordingly, sensitive PDX tumors were characterized by enhanced cell death following treatment with IACS-010759 compared to vehicle, as measured by immunohistochemistry (IHC) analysis of cleaved caspase 3 (Fig. 1l, Supplementary Fig. 5). Thus, consistent with our in vitro data, the in vivo results demonstrated differential response to complex I inhibition, suggesting some cells are endowed with a unique mechanism to overcome OXPHOS depletion.

### Induction of mitochondrial oxidative stress determines sensitivity to complex I inhibition

Inhibition of complex I not only blocks oxygen consumption, but it also generates superoxide, although the mechanism that underlies this phenomenon is not fully understood[19,20]. PDX cell lines exhibited different levels of oxygen consumption rate (OCR), ROS, ATP levels, and mitochondrial mass, irrespectively of their sensitivity (Supplementary Fig. 6a–d). However, whereas complex I inhibition similarly decreased oxygen consumption in all cell lines tested (Fig. 1d–g), exposure to IACS-010759 resulted in a significant accumulation of mitochondrial ROS in sensitive but not in resistant PDX lines, based on analysis using MitoSOX, a fluorogenic dye specifically targeted to the mitochondrial matrix and activated by superoxide in live cells (Fig. 2a, b). Importantly, the induction of mitochondrial ROS preceded cell death induced by IACS-010759 in sensitive PDX lines (Supplementary Fig. 7a–c).

The regulation of ROS homeostasis is a highly compartmentalized process[19]. Although superoxide is produced in the mitochondrial matrix following complex I inhibition[21], it can be transported into the cytoplasm to induce additional oxidative stress. To further evaluate the compartmentalization of ROS following complex I inhibition, we used pCS2-HyPer7, an ultrasensitive, pH-stable GFP probe for hydrogen peroxide ($H_2O_2$). This probe was linked with localization-specific sequences targeting its expression to various cellular compartments: pCS2+MLS-HyPer7 localized in the mitochondrial matrix, and pCS2 + HyPer7-NES localized in the cytoplasm[22]. MitoParaquat (MitoPQ), a mitochondria-targeted redox cycler, and Paraquat (PQ), a cellular ROS inducer, were used as positive controls for mitochondrial ROS and cytoplasmic ROS, respectively (Supplementary Fig. 8). In an IACS-010759-sensitive PDX line, IACS-010759 treatment induced ROS in the mitochondrial matrix, but not in the cytoplasm (Fig. 2c–f), suggesting that regulation of ROS signaling within the mitochondrial matrix may be related to the sensitivity of these cells to complex I inhibition.

ROS production induces cell death because of oxidative stress[19,23]. To determine whether the cell death caused by complex I inhibition was due to the accumulation of mitochondrial ROS, we used MitoPQ to increase ROS levels selectively in the mitochondrial matrix[24]. Combined MitoPQ and IACS-010759 treatment in resistant PDX lines robustly increased cell death (Fig. 2g, Supplementary Fig. 9a). Consistently, decreasing mitochondrial ROS levels using a mitochondria-targeted ROS scavenger, Mito-quinone (MitoQ), in sensitive PDX lines prevented IACS-010759-induced cell death (Fig. 2h). Superoxide dismutases (SODs) protect cells from oxidative damage by transferring superoxide anions ($O_2^{\bullet-}$) to generate $H_2O_2$ and $O_2$[25]. Among the

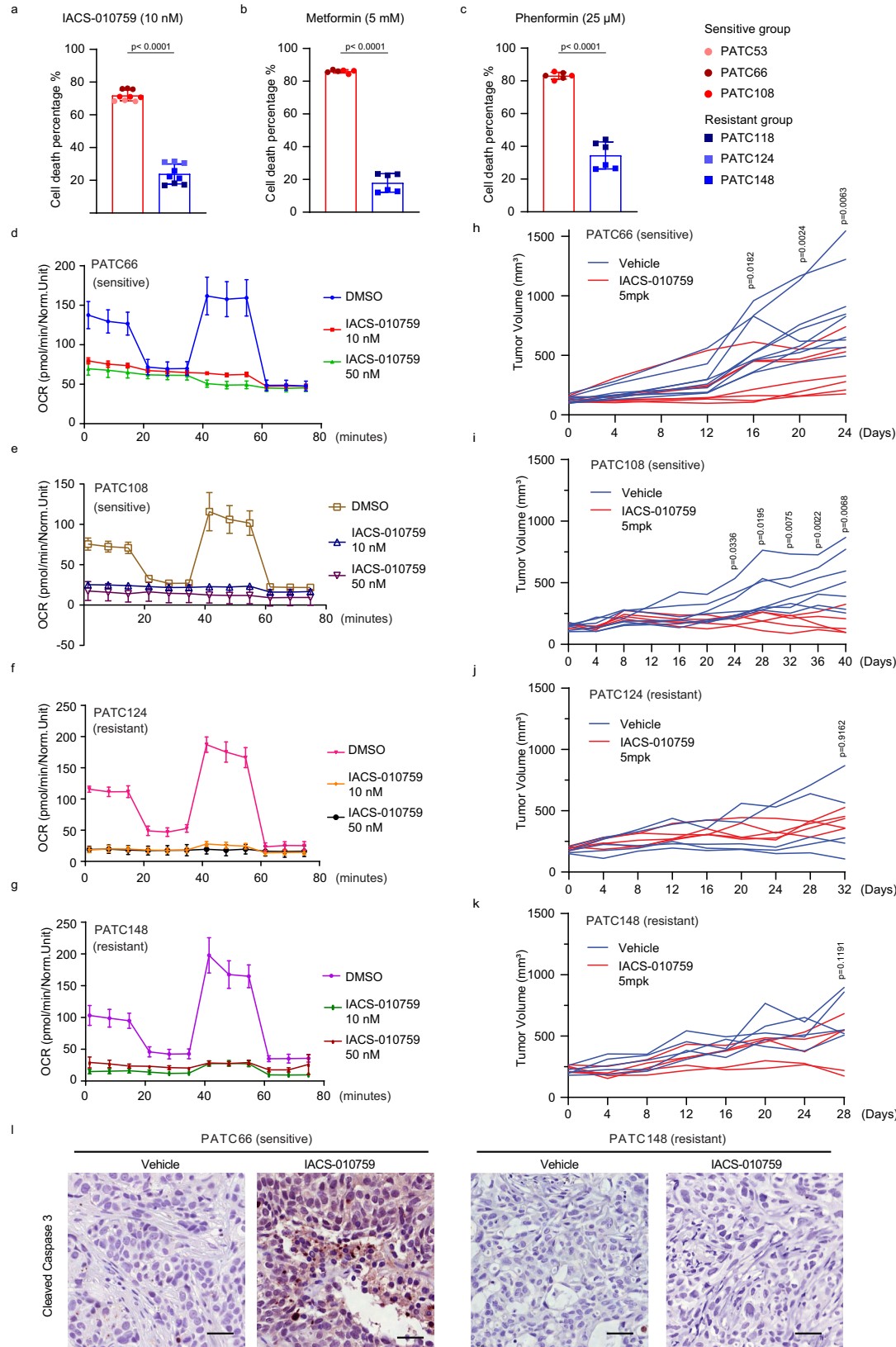

known SODs, manganese-SOD (SOD2) is distributed in the mitochondrial matrix and is primarily responsible for mitochondrial ROS scavenging. We used CRISPR-Cas9 technology to delete *SOD2* in two IACS-010759-resistant PDX lines to study the effect of SODs on sensitivity to complex I inhibition. *SOD2* deletion significantly increased mitochondrial ROS accumulation and cell death following IACS-010759 treatment in both resistant PDX lines (Fig. 2i, j, Supplementary Fig. 9b), further supporting a causal role for mitochondrial ROS in cell death induced by complex I inhibition. Together, our data indicate that the accumulation of mitochondrial ROS is a key determinant of sensitivity to complex I inhibition in PDAC cells.

**Fig. 1 | Identification of latent vulnerability to complex I inhibition in a subset of pancreatic ductal adenocarcinoma patient-derived xenografts. a–c** Cell viability assay separates sensitive (PATC53/66/108) and resistant groups (PATC118/124/148) upon 3 days of treatment with 10 nM IACS-010759 (**a**), 5 mM metformin (**b**), and 25 μM phenformin (**c**), respectively. Three biologically independent replicates per cell line. Three cell lines are classified into sensitive group and other 3 lines as resistant group. Data represent mean ± S.D between groups. **d–g** Oxygen consumption ratio (OCR) in PATC66 (**d**), PATC108 (**e**) and in PATC124 (**f**), PATC148 (**g**) lines, upon 24 h of treatment with IACS-010759 at the indicated concentrations. $N = 3$ biologically replicates per measuement point. Data represent mean ± S.D.

**h–k** Xenograft tumor growth of PDX PATC66 (**h**), PATC108 (**i**), PATC124 (**j**), and PATC148 (**k**) with vehicle or 5 mg/kg IACS-010759. Mice were treated with a fasting/feeding cycle protocol as described in materials and methods. Tumor volume was measured every 4 days for a period of indicated days. $N = 5$–9 mice per group. **l** Representative IHC for cleaved caspase 3 activity in PATC subcutaneous tumors treated with vehicle or 5 mg/kg IACS-010759 on the 23rd (PATC66) and 26th (PATC148) days. The experiment had been repeated individually 3 times with similar results. Scale bar, 50 μm. Statistical analysis by two-tailed Students' unpaired $t$ test with significance indicated (**a–c**, **h–k**). Source data are provided as a Source Data file.

## MUFAs are critical for ROS homeostasis and survival following complex I inhibition

To delineate the mechanism underlying the differential responses to complex I inhibition, we compared cellular metabolic profiles of IACS-010759-sensitive and -resistant PDX lines. Both polar and lipid metabolites were evaluated (Supplementary Data. 1 and 2). While polar metabolomics identified a few metabolites involved in the TCA cycle (citrate, aconitate, and succinate) that were upregulated in the resistant PDX lines (Supplementary Fig. 10a–c), lipid metabolomics revealed a stark difference between the lipid profiles of IACS-010759-sensitive versus -resistant models (Fig. 3a). Levels of monounsaturated fatty acids (MUFAs), especially C18:1, were significantly higher in resistant PDX lines (Fig. 3a, Supplementary Fig. 11a). In contrast, sensitive PDX lines had higher levels of polyunsaturated fatty acids (PUFAs), such as C20:4 and C22:4 (Fig. 3a, Supplementary Fig. 11a).

To elucidate the role of different lipid species in resistance to mitochondrial complex I inhibition, we treated PDX lines with IACS-010759 or DMSO control and supplemented the medium with either BSA or a BSA-conjugated saturated fatty acid (stearic acid, C18:0), a MUFA (oleic acid, C18:1), or a PUFA (α-linolenic acid, C18:3). As shown in Supplementary Fig. 11b, c, oleic acid administration prevented the induction of mitochondrial ROS and of cell death in sensitive PDX lines upon complex I inhibition, while neither stearic acid nor α-linolenic acid had this effect. On the other hand, supplementation with PUFAs (arachidonic acid C20:4, which were abundant in sensitive PDX lines, or mead acid C20:3) increased mitochondrial ROS levels and sensitized resistant PDX cells to complex I inhibition (Supplementary Fig. 11d–h). These observations are consistent with recent reports that abundant PUFAs induce cell death by serving as substrates for lipid peroxidation[26,27], whereas MUFAs decrease lipid peroxidation[28,29].

Stearoyl-CoA desaturase 1 (SCD1) is the key enzyme for MUFA biosynthesis from saturated fatty acids[30]. We thus examined the effect of SCD1 inhibition on mitochondrial ROS and cell survival using a small molecule inhibitor of SCD1, A939572[31]. Resistant PDX cells were treated in vitro with IACS-010759, A939572, or both, and then exposed to MitoSOX to evaluate mitochondrial ROS accumulation. A939572 significantly enhanced mitochondrial ROS in IACS-010759-treated cells compared to cells exposed to either IACS-010759 or A939572 (Fig. 3b). A939572 also dramatically increased sensitivity to mitochondrial complex I inhibition in resistant PDX lines (Fig. 3d). Consistently, combined treatment with IACS-010759 and A939572 resulted in enhanced inhibition of tumor growth in vivo (Fig. 3e).

MUFAs are known scavengers of lipid peroxidation[29]. In addition to mitochondrial ROS, lipid peroxidation was also elevated in IACS-010759-sensitive PDX lines compared to resistant ones following complex I inhibition (Supplementary Fig. 12a, b). We also supplemented the medium of resistant cell lines with or without PUFAs (arachidonic acid C20:4) and treated them with IACS-010759 or DMSO. Consistent with our other findings, PUFA supplementation robustly increased lipid oxidation in IACS-010759-treated resistant PDX lines (Supplementary Fig. 12c). Similarly, blocking the generation of MUFAs by inhibiting SCD1 increased lipid oxidation in resistant cells (Fig. 3c). As lipid peroxidation is correlated with ferroptosis[28], treatment with various cell death inhibitors was evaluated to further dissect the cell

death mechanism induced by IACS-010759. Specifically, we tested the pan-caspase inhibitor, Z-VAD-FMK; the ferroptosis inhibitors, ferrostatin-1, liproxstatin-1, and deferoxamine (DFO); and the necrosis inhibitor, necrostation-1. Among them, Z-VAD-FMK and DFO, but not necrostatin-1, ferrostatin-1 or liproxstatin-1, partially rescued cell death induced by IACS-010759 (Supplementary Fig. 13a, d), indicating that the cell death induced by complex I inhibition may involve both apoptosis and ferroptosis. These results suggest that MUFAs directly contribute to modulating sensitivity to complex I inhibition through the regulation of mitochondrial ROS and lipid peroxidation.

## Mitochondrial MUFA-linked ether phospholipids modulate sensitivity to complex I inhibition

Our data indicate that regulation of redox homeostasis specifically within the mitochondria confers resistance to complex I inhibition. To examine how lipid composition may contribute to the regulation of ROS in the mitochondria, we next conducted a lipidomic analysis of mitochondria isolated from sensitive or resistant PDAC cell lines. Similar to our findings from whole-cell lipid profiling, we detected an elevation of MUFAs and a decrease in PUFAs in mitochondria in resistant compared to sensitive PDX lines (Fig. 3f, Supplementary Data. 3). Interestingly, most of the lipid species enriched in the mitochondria of resistant cells were ether phospholipids, and of these C18 MUFA-linked ether phospholipids (O-C18:1) were heavily enriched (Fig. 3f). Ether lipids are important components of biological membranes[32,33]. Their de novo synthesis is initiated in peroxisomes and completed in the endoplasmic reticulum (ER), and they are then transferred to mitochondria and other organelles[33,34]. Recent reports demonstrated that ether phospholipids are involved in adaptation to hypoxia and in ferroptosis[35–37]. To examine the potential role of ether lipids in resistance to mitochondrial complex I inhibition, we used CRISPR-Cas9 technology to knock out two key peroxisomal enzymes required for ether lipid biosynthesis, glyceronephosphate O-acyltransferase (*GNPAT*) and alkylglycerone phosphate synthase (*AGPS*), in resistant PDX lines (Supplementary Fig. 14a)[33]. Deletion of *GNPAT* or *AGPS* specifically depleted C18:1-linked ether phospholipids without significantly impacting the total amount of C18:1-linked phospholipids (Supplementary Fig. 14b, c). Importantly, in resistant PDX lines exposed to IACS-010759, *GNPAT* or *AGPS* depletion led to the accumulation of mitochondrial ROS and lipid peroxidation following complex I inhibition, accompanied by robust induction of cell death (Fig. 3g–i), but these effects were not observed in vehicle-treated cells. These findings were recapitulated in cells carrying a deletion of another peroxisomal enzyme involved in ether lipid synthesis, fatty acyl-CoA reductase 1 (*FAR1*) (Fig. 3g–i, Supplementary Fig. 14a). Furthermore, in *GNPAT*-deleted cells exposed to IACS-010759, supplementation of medium with C18:1-linked ether lipids (O-C16-18:1 PC), which is enriched in mitochondria (Supplementary Fig. 15), abrogated the induction of mitochondrial ROS and lipid peroxidation and prevented cell death (Fig. 4a–c). Interestingly, there were no significant differences in the expression of ether lipid biosynthesis enzymes or in peroxisomal mass between sensitive and resistant PDX

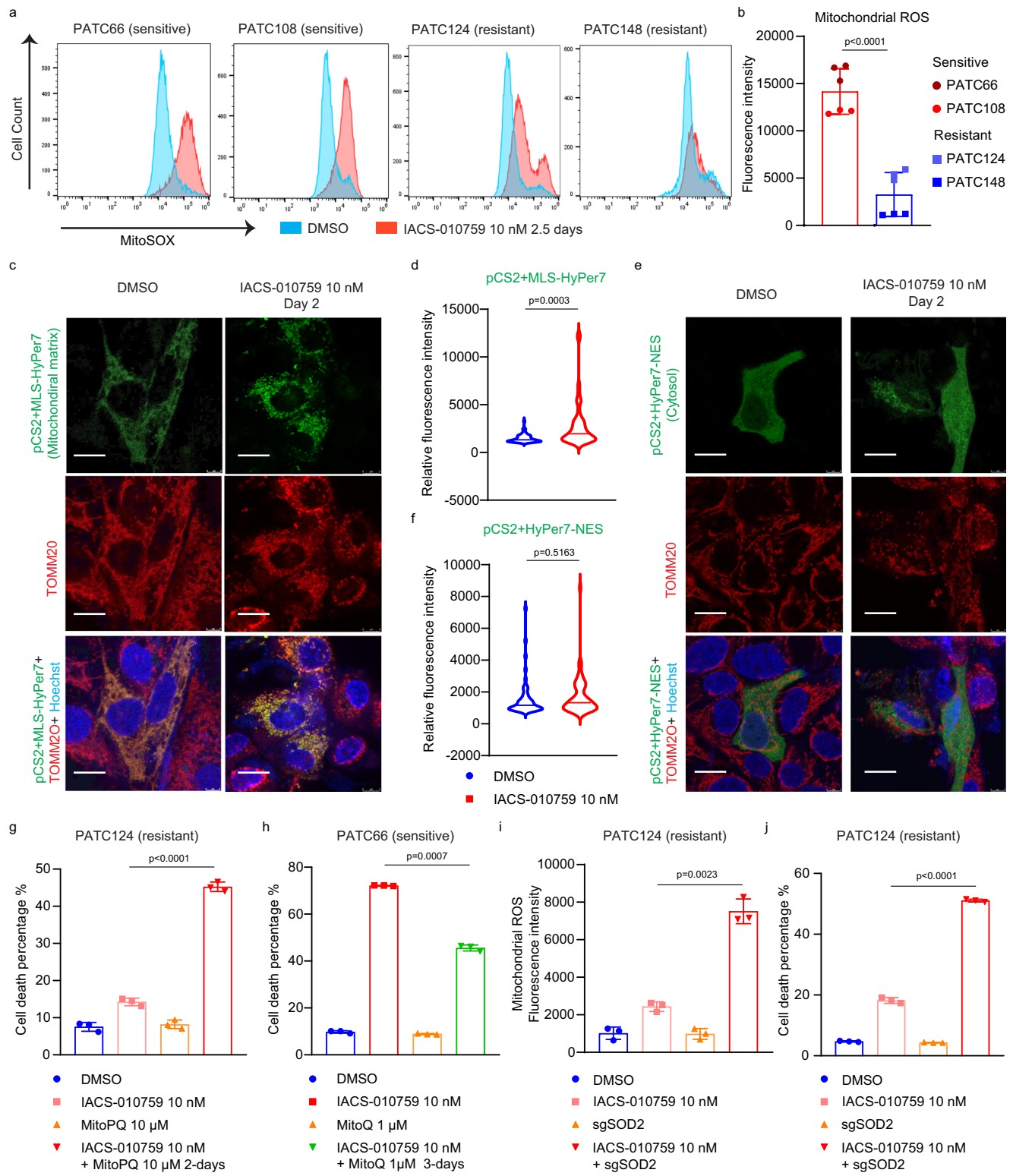

lines (Supplementary Fig. 16a). However, isotopy flux tracing with $^{13}C_{18}$-oleic acid showed enhanced $^{13}C$ labeled-oleic acid flux into ether lipids (PC 16:0 O/18:1) in resistant PDX lines compared to sensitive PDX lines, and this flux into ether lipids could be blocked by depleting GNPAT (Supplementary Fig. 16b–e). Importantly, exogenous oleic acid was unable to rescue the ROS accumulation and cell death phenotypes induced by IACS-010759 in *GNPAT*-deficient cells (Fig. 4d, e), indicating that MUFA-mediated survival and inhibition of mitochondrial ROS upon complex I inhibition

depend on the conjugation of MUFAs to ether lipids. In subcutaneous xenograft models of resistant PDX lines, *GNPAT* deletion alone did not impede tumor growth; however, treatment of animals harboring *GNPAT*-deleted tumors with IACS-010759 significantly suppressed tumor growth and increased cell death (Fig. 4f, g, Supplementary Fig. 17a, b), underscoring the importance of ether lipids in resistance to complex I inhibition.

To evaluate whether medium supplementation with different fatty acid-containing ether lipids would contribute to ROS

**Fig. 2 | Induction of mitochondrial oxidative stress enhances sensitivity to complex I inhibition. a** Mitochondrial ROS were measured by MitoSOX staining and flow cytometry in the indicated cells grown for 60 hr in DMSO or 10 nM IACS-010759. The experiment was repeated independently three times and representative results were shown. **b** Quantification of mitochondrial ROS (MFI comparison) in sensitive (PATC66/108) and resistant (PATC124/148) groups after 60 hr treatment with 10 nM IACS-010759. Three biologically independent replicates per cell line. Data represent mean ± S.D between the sensitive group (2 cell lines) and resistant group (2 cell lines). **c–f** Cells were transfected with the hydrogen peroxide indicator pCS2+MLS-Hyper7 (mitochondrial matrix-targeted) (**c**) and pCS2 + HyPer7-NES (cytoplasm-targeted) (**e**), and treated with 10 nM IACS-010759 at the indicated times. Mitochondria were defined by staining with an antibody against the mitochondrial outer membrane protein TOMM20. Nuclei were stained with Hoechst 33578. Scale bar, 10µm. **d** Quantification of fluorescence intensity in (**c**). **f** Quantification of fluorescence intensity in (**e**). For (**d**) and (**f**), at least 50 cells with positive green fluorescence from three biologically independent replicates were calculated for each group. **g** Cell death was detected by propidium iodide staining and flow cytometry in PATC124 cells treated with DMSO or 10 nM IACS-010759 for 2 days, in the presence or absence of the mitochondrial-ROS inducer MitoPQ (10 µM). Data represent mean ± S.D of three biologically independent replicates. **h** Cell death in PATC66 cells treated with DMSO or 10 nM IACS-010759 for 3 days, in the presence or absence of the 1 µM mitochondrial ROS scavenger, MitoQ. Data represent mean ± S.D of three biologically independent replicates. **i, j** Mitochondrial ROS (**i**) and cell viability (**j**) were detected by flow cytometry in PATC124 cells infected with control sgRNA (sgCTRL) or sgRNA targeting SOD2 (sgSOD2) following treatment with 10 nM IACS-010759 for 3 days. Data represent mean ± S.D of three biologically independent replicates. Statistical analysis by two-tailed Students' unpaired *t* test with significance indicated (**b, d, f, g–j**). Source data are provided as a Source Data file.

regulation and cell survival, we grew sensitive PDX lines in a medium supplemented with exogenous ether phospholipids linked with SFAs (O-C16-04:0 PC), MUFAs (O-C16-18:1 PC), or PUFAs (O-C16-20:3 PC) and treated them with IACS-010759 or DMSO (Supplementary Fig. 18a). While SFA- and PUFA-linked ether lipids increased IACS-010759-induced cell death (Supplementary Fig. 18b), supplementation with MUFA-linked ether lipids (O-C16-18:1 PC) significantly reduced mitochondrial ROS and lipid peroxidation following IACS-010759 treatment (Fig. 4h, i, Supplementary Fig. 18c) and rescued complex I inhibition-induced cell death (Fig. 4j, k). Repeating these experiments with phenformin resulted in outcomes similar to what was observed with IACS-010759 (Supplementary Fig. 19). These findings further demonstrate the importance of both the phospholipid ether linkage and the conjugated MUFA chain for the regulation of mitochondrial ROS homeostasis.

## MUFA-linked ether phospholipids promote assembly of mitochondrial supercomplexes

The regulation of both mitochondrial superoxide and lipid peroxidation by MUFA-linked ether lipids suggests that in addition to their function as scavengers of lipid peroxidation, these molecules may also regulate mitochondrial ROS production. Mitochondria in cells can enhance ETC efficiency and decrease electron leakage and ROS accumulation by forming supercomplexes[38–40], and it has been established that membrane lipid composition contributes to the assembly and stability of these supercomplexes[41,42]. To evaluate this in our models, we used blue native PAGE, a method for identifying mitochondrial supercomplexes (Supplementary Fig. 20a). Interestingly, high-molecular-weight supercomplexes (hmwSCs) were more abundant and active in resistant compared to sensitive PDX lines (Fig. 5a, b and Supplementary Fig. 20b, 21a). Further, supplementation with MUFA-linked ether lipids (O-C16-18:1) increased the abundance of hmwSCs in sensitive PDX lines cells (Fig. 5c, d, Supplementary Fig. 21b), while supplementation of PUFA-linked ether lipids (O-C16-20:3) had minimal impact on the amount of hmwSCs in these cells (Fig. 5e, f, Supplementary Fig. 21c). Moreover, deletion of *GNPAT* or *AGPS* in resistant cells decreased assembly of hmwSCs, and this effect was rescued by supplementation with MUFA-linked ether lipids, but not PUFA-linked ones (Fig. 5g, h, Supplementary Fig. 21d). To further verify the impact of hmwSC assembly on sensitivity to complex I inhibition, we knocked out ubiquinol-cytochrome C reductase complex assembly factor 3 (*UQCC3*), a supercomplex assembly factor[43], to abolish hmwSC formation (Supplementary Fig. 22a). *UQCC3* deletion significantly increased sensitivity to complex I inhibitor treatment in resistant PDX lines (Supplementary Fig. 22b), underscoring the importance of hmwSC assembly for resistance to complex I inhibition. Together, our findings reveal a previously unrecognized function of MUFA-linked ether lipids in the assembly of mitochondrial supercomplexes, which may influence cell sensitivity to inhibition of ETC complex I.

## Discussion

In this study, we demonstrate that mitochondrial ROS homeostasis is essential for the survival of PDX-derived pancreatic cancer cells with impaired function of mitochondrial complex I, and that peroxisome-derived ether phospholipids, especially those conjugated with MUFAs, play a critical role in maintaining mitochondrial redox balance in this context. Inhibition of ether lipid biosynthesis synergizes with mitochondrial complex I inhibition to induce mitochondrial ROS and inhibit tumor growth. We also discovered a function of MUFA-linked ether lipids to regulate the formation of mitochondrial ETC hmwSC formation that may be relevant to this synergy between complex I function and MUFA-linked ether lipids (Fig. 5i).

Enhanced anabolic metabolism and redox homeostasis are key features of tumor metabolism. As the central hub for these metabolic programs, the role of mitochondrial function in tumor development and therapy resistance is increasingly appreciated, leading to concerted efforts to evaluate the potential of targeting mitochondrial metabolism for cancer therapeutics. There is a clear need to investigate the interactions between mitochondria and other cellular metabolism programs more thoroughly, as this may shed light on possible therapeutic targeting strategies.

A growing body of evidence demonstrates multiple roles for lipid metabolism in cancer progression, proliferation, and therapy response[36,44–46]. The divergent functions of different lipid species, such as SFAs, MUFAs, and PUFAs, are increasingly defined. Our study revealed that MUFAs are accumulated in a subset of PDAC cells and have a critical function to regulate mitochondrial ROS. Blockade of MUFA synthesis by inhibition of SCD cooperates with complex I inhibition to induce mitochondrial ROS and cell death. This is consistent with a recent study that demonstrated that low-glycemic diets inhibit PDAC tumor growth by suppressing the activity of SCD and decreasing MUFA levels in tumor cells[47].

MUFAs have long been recognized to function as scavengers of lipid peroxidation[29]. Interestingly, our data suggest that the effect of MUFAs on mitochondrial ROS and lipid peroxidation is likely due to their conjugation to ether lipids. Ether lipids are peroxisome-generated glycerophospholipids where the hydrocarbon chain at the sn-1 position is linked to the glycerol backbone through an ether bond, instead of to the ester bond more commonly found in diacyl phospholipids[33]. Despite their discovery almost a century ago[48], the biological functions of ether lipids, especially their roles in human cancers, remain poorly understood, although the level of ether lipids seems to be higher in tumor compared to normal cells[33]. Recently, genetic screens have identified that ether lipids regulate lipid peroxidation and ferroptosis[36,49]. However, rather than a protective role of ether lipids against lipid peroxidation, it was shown

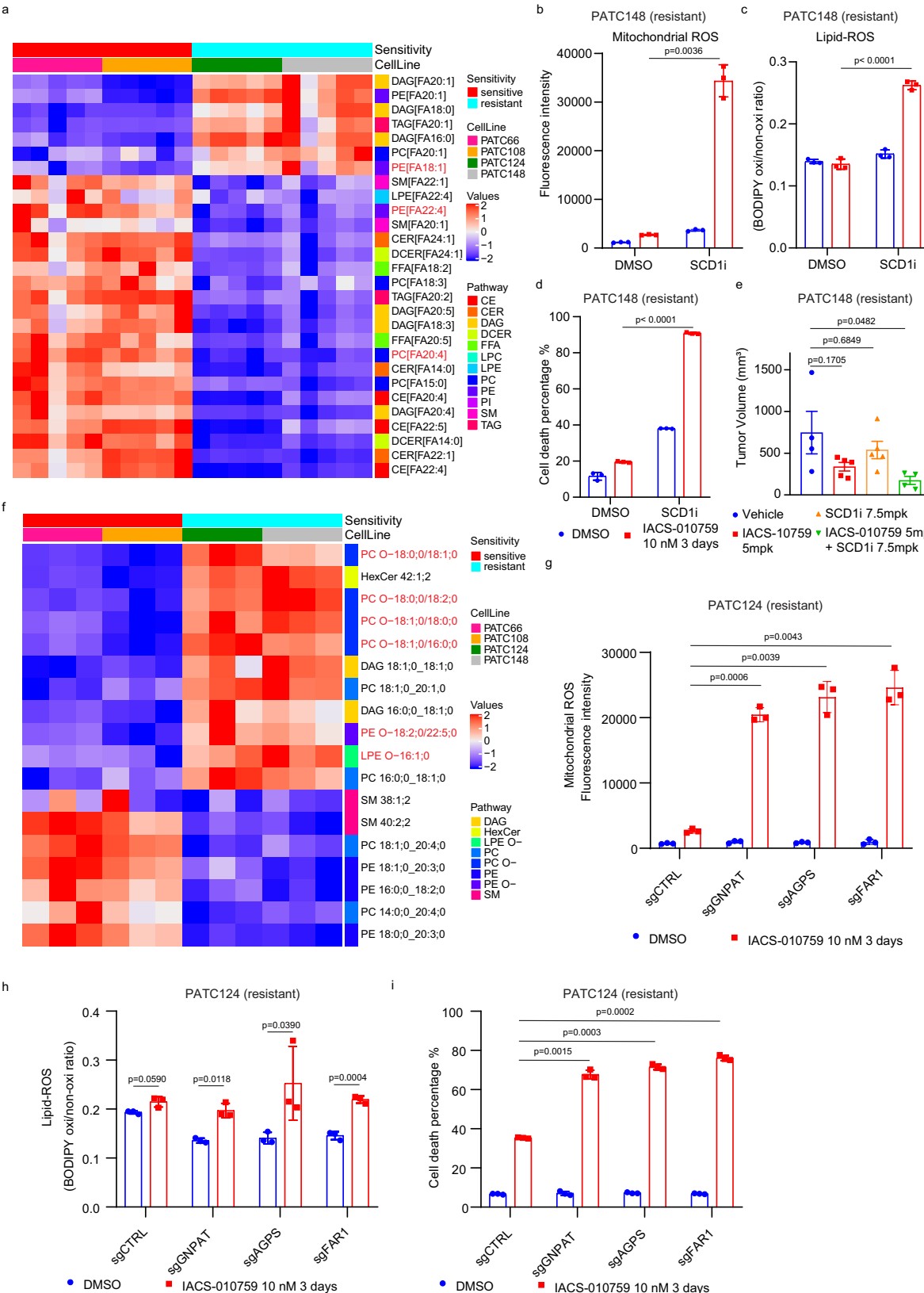

that blocking peroxisome ether lipid biosynthesis rescued lipid peroxidation-induced ferroptosis in a variety of normal and tumor cells[36,49]. This is likely because ether lipids are generally conjugated with PUFAs, which serve as the major substrates for lipid peroxidation[49]. In contrast, most ether lipids in PDAC cells are linked with MUFAs, which are known scavengers of lipid

peroxidation[29]. Moreover, the structural features of the ether lipid backbone may also function as a scavenger of lipid peroxidation. Plasmalogens, a major form of ether lipid, contain the vinyl ether linkage at the sn-1 position, which is a cis double bond adjacent to the ether linkage. The vinyl ether bond is particularly susceptible to oxidative damage and, therefore, plasmalogens may function as

**Fig. 3 | MUFAs-linked ether phospholipids modulate sensitivity to complex I inhibition. a** Heatmap of the lipid metabolites detected at significantly different levels in the treatment-sensitive cells (PATC66/108) compared to resistant ones (PATC124/148). Metabolites with $P < 0.05$ between the two groups were listed. $N = 5$ for each cell line. **b–d** Mitochondrial-ROS (**b**), lipid-ROS (**c**) and cell death (**d**) as measured with MitoSOX, BODIPY 581/591 C11, and propidium iodide, respectively in PATC148 cells upon 3 days of treatment with DMSO or 10 nM IACS-010759, in the presence or absence of the SCD1 inhibitor A939572 (10 μM). Data represent mean ± S.D of three biologically independent replicates. **e** PATC148 xenograft tumor growth in mice treated with vehicle or 5 mg/kg IACS-010759, once every other day, alone or in combination with the SCD1 inhibitor A939572 (7.8 mg/kg, once every other day). Mice were treated with fasting/feeding cycle protocol. Tumor measurements were made on the 33$^{rd}$ day of experiment, 9 hr after the last treatment. $N = 4$ for vehicle and combination groups; $N = 5$ for IACS-010759 and A939572 groups. Data represent mean ± SEM. Statistical analysis by ordinary one-way ANOVA followed by Tukey's multiple comparisons test. **f** Heatmap of lipid species detected in purified mitochondria from sensitive group (PATC66/108) and resistant group (PATC124/148). Metabolites with $P < 0.05$ (by two-tailed Students' unpaired $t$ test) between the two groups were listed. $N = 3$ for each cell line. **g–i** Mitochondrial-ROS (**g**), lipid-ROS (**h**), and cell death (**i**) as measured with MitoSOX, BODIPY 581/591 C11, and propidium iodide, respectively in PATC124 cells infected with control sgRNA (sgCTRL) or sgRNA targeting GNPAT, AGPS, and FAR1, and treated for 3 days with DMSO or 10 nM IACS-010759. Data represent mean ± S.D of three biologically independent replicates. Statistical analysis by two-tailed Students' unpaired $t$ test with significance indicated (**b–d**, **g–i**). Source data are provided as a Source Data file.

scavengers of lipid peroxidation to protect the PUFA moieties of membrane lipids from oxidation[50]. Therefore, both the ether lipid backbone and its conjugated fatty acid species can determine the role of ether lipids against lipid peroxidation.

In addition to its regulation of lipid peroxidation, our study discovered that ether lipids may also control the production of superoxide by promoting the formation of mitochondrial hmwSCs. It is well established that the formation of hmsSCs enhances the efficiency of OXPHOS and reduces the electron leakage that generates superoxide as a toxic byproduct[38–40]. Pancreas tumor cells grow in a severely hypoxic milieu[51], which makes them more susceptible to ROS during rapid proliferation. One strategy used by PDAC cells to prevent oxidative damage is to form hmwSCs[52]. It is possible that the increased formation of hmwSCs in PDAC cells is made possible due to the enrichment of ether lipids in their mitochondrial membrane. Interestingly, it has been recently demonstrated that deletion of peroxisomal genes, such as *PEX10* and *GNPAT*, increases cell growth inhibition in low-oxygen conditions[53], further supporting the role of ether lipids in adaptation to hypoxic environments. It will be important to further explore whether ether lipid biosynthesis may represent a unique vulnerability for pancreas cancer cells.

## Methods

### Cell lines culturing and treatment

Human PATC cell lines (PATC53, PATC66, PATC108, PATC118, PATC124, and PATC148) were generated from pancreatic duct adenocarcinoma by patient-derived xenograft (PDX) models as previously described[9]. Cells were maintained and treated in DMEM/low glucose (Cytiva HyClone) medium supplemented with 10% heat-inactivated fetal bovine serum (FBS, Gibco) and 1% Penicillin/Streptomycin (Cytiva HyClone) antibiotics. All PATC cell lines were cultured in standard incubation conditions (5% $CO_2$, 37 °C) and treated with compounds at early passages (within 20 passages).

Complex I inhibitor IACS-010759 was synthesized and kindly provided by the Institute for Applied Cancer Science (IACS), the University of Texas MD Anderson Cancer Center (MDACC)[13]. Metformin hydrochloride (#PHR1084), phenformin hydrochloride (#PHR1573), oligomycin A (#75351), bovine serum albumin fraction V (#03117057001), oleic acid-albumin from bovine serum (O3008), and stearic acid (FA 18:0, #85679) were purchased from Sigma-Aldrich. Antimycin A (#ab141904) was ordered from Abcam. MitoPQ (#18808), MitoQ (#89950), oleic acid (FA 18:1, #24659), α-linolenic acid (FA 18:3, #21910), arachidonic acid (FA 20:4, #90010) were obtained from Cayman Chemical Company. Ether phospholipids 1-O-hexadecyl-2-oleoyl-sn-glycero-3-phosphocholine (O-C16-18:1 PC, #878112), 1-O-hexadecyl-2-(8Z,11Z,14Z-eicosatrienoyl)-sn-glycero-3-phosphocholine (O-C16-20:3 PC, #878122), and 1-O-hexadecyl-2-butyryl-sn-glycero-3-phosphocholine (O-C16-04:0 PC, #878115) were provided by Avanti Polar Lipids.

### Generation of CRISPR/Cas9 knockout cell lines

The sgRNAs targeting SOD2 / GNPAT / AGPS / FAR1/ UQCC3 were designed using CRISPick website software (https://portals.broadinstitute.org/gppx/crispick/public) supported by BROAD Institute. The sequences are SOD2, ACAAACCTCAGCCCTAACGG; GNPAT, ATGGCTAAAAGGCTTAACCC; AGPS, TTTGTTCAAATACGGTCAGT; FAR1, AGCACTAATCCTTTCCACTG; UQCC3, #1 CTCCCGGGGTCACGATAACG, #2 GCGGAAGCAGGAAATGCTAA. The sgRNAs were individually inserted into LentiCRISPR v2 one vector system (Addgene, Plasmid #52961) and transfected in HEK-293 cells (CRL-1573™) for 48 hr. The culturing medium was isolated and filtered (0.22 μm pore size, MilliporeSigma™ SCGP00525). After ultracentrifuge at 25,000 g for 2 hr, virus pellets were suspended by PBS and infected into PATC lines. The stable CRISPR/Cas9 knockout PATC lines targeting indicated genes were generated after puromycin (1 mg/mL, Invivogen, # ant-pr-1) selection. Western blotting was detected for the loss of proteins.

### Cell sensitivity studies

**Cell death assay.** A total of $3 \times 10^5$ cells per well were seeded in 6-well plates and treated with indicated concentrations of compounds for specific durations. Cells were harvested, stained with propidium iodide (PI, BD-Pharminogen#51-66211E), and detected within 1 hr of staining by using a Gallios flow cytometer (Beckman Coulter, Inc. Brea, CA) and following the manufacturer's instructions. Data were analyzed by Kaluza and Flowjo (Version 10.6.2) software.

**Crystal violet stain.** A total of $1 \times 10^5$ cells were plated and treated with 10 nM IACS-010759 or with vehicle for 5 days. Cells were fixed and stained with a 1% crystal violet solution that included 20% methanol. After 1 hr of staining at room temperature, cells were thoroughly washed with PBS, dried, and scanned with Epson Perfection V700 Photo.

**Bright field images.** A total of $3 \times 10^5$ cells per well were cultured in 6-well plates and treated with indicated concentrations of compounds for specific durations. Cells were captured and analyzed by Olympus microscopy.

### Mitochondrial oxygen consumption rate (OCR) detection

A total of $5 \times 10^3$ cells were seeded with 80 μL Dulbecco's Modified Eagle Medium (DMEM) in 96-well Seahorse XFe96 V3 PS Cell culture microplates (#101085-004) and treated with indicated concentrations of IACS-010759 or vehicle for 24 hr. Cells were incubated in Seahorse XF base medium supplemented with 5.5 mM D-glucose, 2 mM glutamine, and 1 mM pyruvate. Oxygen consumption rate (OCR) was then measured by Seahorse XFe96 Analyzer (Agilent). Seahorse OCR analysis for PATC cell lines were performed according to Seahorse Biosciences instructions from Seahorse XF Cell Mito Stress Test Kit (#103015-100). Oligomycin (0.5 μM), FCCP

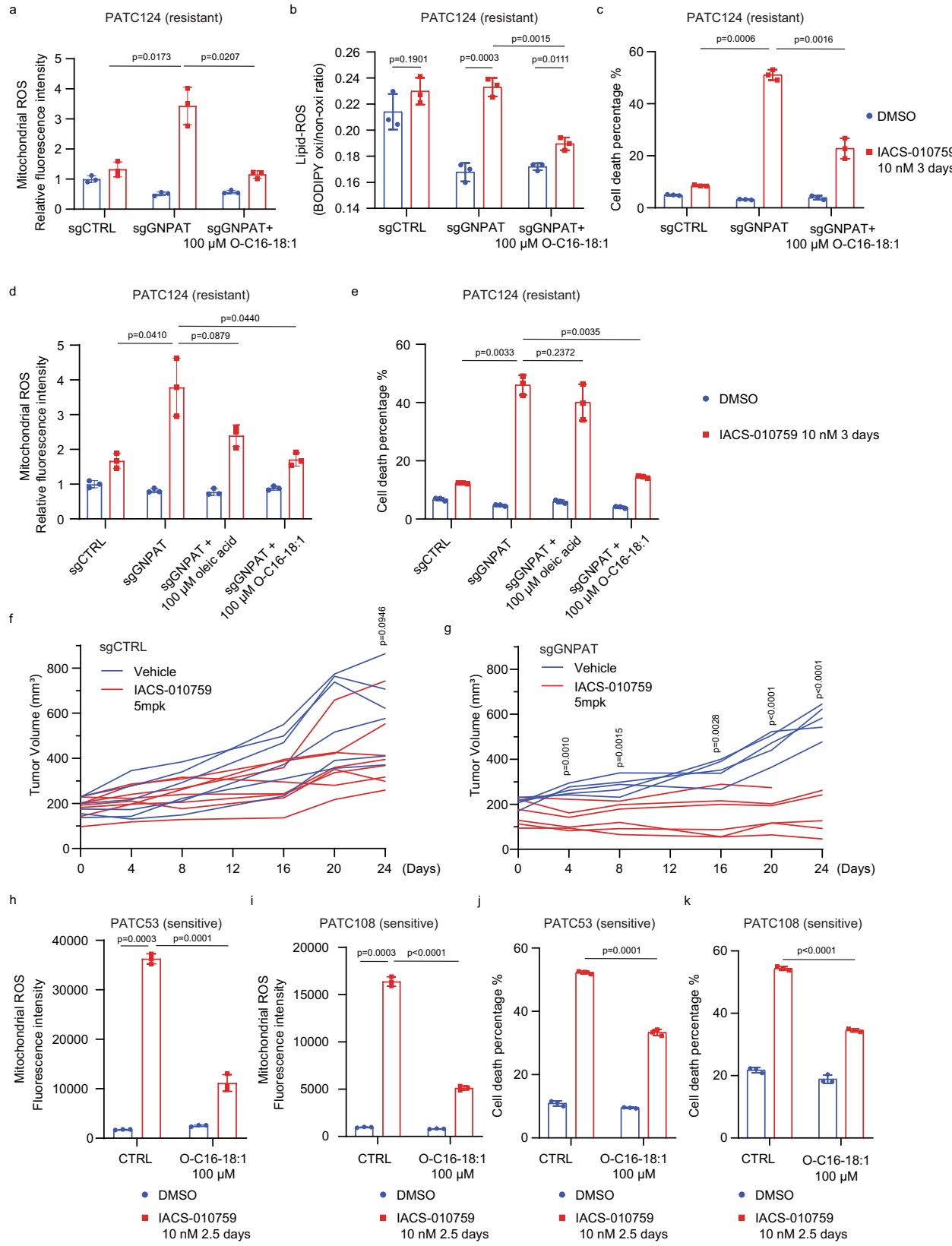

(optimized based on cell lines, 0.125-0.25 μM), and rotenone/antimycin A (0.5 μM) were injected into ports and measured by XFe96 sensor cartridges (#102416-100) consecutively. Data were normalized by Hoechst stain-positive cell numbers obtained by the Operetta CLS High-Content Analysis System (PerkinElmer).

**Metabolic stress assays**
**Mitochondrial reactive oxygen species (ROS) and lipid peroxidation detection.** A total of $3 \times 10^5$ PATC cells were plated and treated with indicated compounds for specified durations. Cells were harvested, washed with cold PBS, and stained with 5 μM MitoSOX (Life Technologies, #M36008) or 1 μM BODIPY 581/591 C11 (Thermo Fisher

**Fig. 4 | Exogenous MUFA-linked ether phospholipids promote resistance to mitochondrial complex I inhibition. a–c** Mitochondrial ROS (**a**), lipid peroxidation (**b**), and cell death (**c**) as measured with MitoSOX, BODIPY 581/591 C11, and propidium iodide, respectively in PATC124 cells infected with control sgRNA (sgCTRL) or sgRNA targeting GNPAT (sgGNPAT) in the presence or absence of 100 µM O-C16-18:1 PC. Data represent mean ± S.D of 3 biologically independent replicates. **d–e** Mitochondrial ROS level (**d**) and cell death (**e**) in sgCTRL- and sgGNPAT-PATC124 cells, in the presence of 100 µM oleic acid or 100 µM O-C16-18:1 PC. Data represent mean ± S.D of three biologically independent replicates. **f, g** Growth curves of sub-cutaneous xenograft tumors derived from PATC148 cells infected with control sgRNA (sgCTRL) or sgRNA targeting GNPAT (sgGNPAT) and treated

with vehicle or 5 mg/kg IACS-010759. Tumor volume was measured every 3–5 days for 24 days. $N = 5$–8 per group. **h, i** Mitochondrial ROS were detected with MitoSOX upon treatment with DMSO or 10 nM IACS-010759 in PATC 53 (**h**) and PATC108 (**i**) cells grown in the presence or absence of 100 µM ether-MUFA (O-C16-18:1 PC). Data represent mean ± S.D of three biologically independent replicates. **j, k** Cell viability was detected by propidium iodide staining after treatment with DMSO or 10 nM IACS-010759 in PATC53 (**j**) and PATC108 (**k**) cells grown in the presence or absence of 100 µM O-C16-18:1 PC. Data represent mean ± S.D of 3 biologically independent replicates. Statistical analysis by two-tailed Students' unpaired $t$ test with significance indicated (**b–d**, **g–i**). Source data are provided as a Source Data file.

Scientific #D3861) for mitochondrial ROS detection or lipid ROS detection, respectively, for 20 min at 37 °C. Cells were then washed with cold PBS twice and filtered into single-cells suspensions. Cells stained for MitoSOX were then detected by a PE-Texas red filter and separated by flow cytometry. The PE-Texas Red filter was also used for detecting reduced BODIPY-C11, and FITC was used for detecting oxidized BODIPY-C11. Approximately 10,000–20,000 cells were collected and analyzed with Kaluza and Flowjo (Version 10.6.2) software. Peak shifts detected by MitoSOX-based flow cytometry indicated the medium fluorescence intensity (MFI) of mitochondrial ROS production. Further, the ratio oxidized/reduced BODIPY-C11 indicated levels of lipid peroxidation in each group.

**Expression of $H_2O_2$ probes and indication in subcellular scale.** Ultrasensitive $H_2O_2$ probes pCS2+MLS-HyPer7 and pCS2 + HyPer7-NES were gifted by Dr. Vsevolod Belousov (Addgene plasmids, RRID: Addgene_136470; RRID: Addgene_136467). For plasmid transfection, $3 \times 10^5$ PATC66 cells were seeded in 6-well plates for 24 hr. Cells were transfected with pCS2+ plasmids using Lipofectamine 3000 reagent (Life Technologies, #L3000015) according to the manufacturer's protocol. After 24 hr, transfected cells were reseeded onto Nunc TM lab-Tek TM II 8-well chambered cover glass (Thermo Scientific, #155360) and treated with IACS-010759, MitoPQ, or Paraquat (PQ) with specified durations. Cells were then fixed with 4% paraformaldehyde (Boston Bioproducts Inc. BM155500ML) and treated with blocking/permeabilization buffer containing 0.2% Triton X-100 and 1% BSA for 10 min. Afterwards, each well was treated with anti-TOMM20 primary antibody (Proteintech, #11802-1-AP, 1:2000 dilution in PBS) and incubated overnight at 4 °C. Cells were then washed three times with PBST (0.05% Triton X-100) and incubated with a goat polyclonal secondary antibody (Goat anti-Rabbit Alexa Fluor 555, Fisher Scientific, #PIA32732) and Hoechst 33342 nucleic acid stain (Life Technologies, #H1399) for 40 min at room temperature. After washing with cold PBS, samples were covered with mounting medium (VECTOR LABS, #H-1200). Cell imaging was performed by a Leica SP8 Laser scanning confocal microscope and wide-field Nikon Eclipse-Ni microscope with a Hamamatsu C11440 digital camera.

Automatic image segmentation and quantification were performed using a MATLAB 2020b (The MathWorks, Inc.). Otsu's thresholding method and marker-controlled watershed algorithm were employed to segment and specify DAPI-positive nuclear regions of individual cells. Expression levels of NES and MLS in the individual segmented area were then quantified as pixel intensity of green fluorescence. At least 50 cells with positive green fluorescence were then randomly selected for further statistical analysis. The quantified data were plotted in a violin graph using the fluorescent intensity level. Statistical significance was assessed using the Kruskal-Wallis test and the Dunn pairwise test.

### Immunoblotting assays
**Western blotting and antibodies.** After treatment of compounds or knocked-out genes, PATC cells were harvested by digestion with 0.25% trypsin (Corning, #MT25053CI) and washed twice with ice-cold PBS. Cells were then lysed with RIPA lysis buffer (Fisher Scientific, #NC9193720) and

a protease and phosphatase inhibitor cocktail (Life Technologies, #78447) on ice for 30 min, and centrifuged at 20,000 $g$ for 15 min. Supernatants were then transferred into new EP tubes, and the total protein concentration was quantified by a BCA protein assay kit (Thermo Scientific, #23227). Samples were then diluted 6× by using Laemmli loading buffer (Fisher Scientific, #NC9140746) and heated to 95 °C for 5 min. Sodium dodecyl sulfate–polyacrylamide gel electrophoresis (SDS-PAGE) was performed in 4-15% precast polyacrylamide gels (Bio-rad, #4561085). Samples were transferred onto a nitrocellulose membrane (Bio-rad, #1704158). After blocking with 5% BSA or milk in PBST (including 0.1% Tween 20), membranes were incubated with indicated primary antibodies overnight at 4 °C. Samples were then washed with PBST and incubated with indicated secondary antibodies for 1 hr at room temperature and washed with PBST again. Immunoblotting images were developed using a Pierce ECL western blotting substrate kit (VWR, PI32209). The following antibodies were used: anti-Cleaved Caspase 3 (Cell Signaling Technology, #9664 L,1:1000 dilution), anti-Ki67 (Life Technologies, #MA514520,1:200 dilution), anti-SOD2/MnSOD (Abcam, #ab13533,1:2000 dilution), anti-GNPAT (Proteintech, #14931-1-AP, 1:1000 dilution), anti-AGPS (Sigma-Aldrich, #HPA030211, 1:1000 dilution), anti-FAR1 (Novus Biologicals, #NBP1-89847, 1:1000 dilution), total OXPHOS human WB antibody cocktail (Abcam, #ab110411, 1:1000 dilution), anti-ATP5A (Abcam, #ab14748, 1:2000 dilution), anti-vinculin (EMD Millipore, #05-386, clone V284, 1:3000 dilution).

**Mitochondria isolation and Blue Native western blotting.** Mitochondria were isolated using a Mitochondria/Cytosol Fractionation Kit (BioVision, #K256) according to the manufacturer's instruments. Briefly, $1$–$5 \times 10^7$ PATC cells were harvested and washed with ice-cold PBS. Cells were resuspended by 1 mL 1× cytosol extraction buffer (with DTT and protease inhibitors) and lysed on ice for 10 min. Cells were homogenized on ice (40-50 strokes) and centrifuged at 700 $g$ for 10 min at 4 °C. Supernatant was transferred to a 1.5 mL tube and centrifuged at 10,000 $g$ for 30 min at 4 °C. Mitochondria fraction pellets were resuspended with 100 µL ice-cold PBS. Mitochondria quantification was performed by the BCA protein assay kit (ThermoFisher, #23227) according to the manufacturer's instruments. Fifty micrograms of mitochondria pellets were isolated by centrifuge at 10,500 $g$ for 15 min at 4 °C. Mitochondria samples were then immediately stored at −80 °C, or were immediately used to perform Blue Native polyacrylamide gel electrophoresis (BN-PAGE) for detecting mitochondrial supercomplexes. We follow the BN-PAGE and western blotting protocol described by Jha et al.[54]. Briefly, pellets were mixed with 20 µL NativePAGE sample buffer cocktail (Thermo Scientific #BN2003 for 4×NativePAGE sample buffer, BioVision #2082-1 for EZSolution Digitonin, digitonin/protein ratio is 8 g/g) on ice for 20 min and centrifuged at 20,000 $g$ 10 min at 4 °C. Fifteen microlitres of supernatant was mixed with 2 µL Coomassie G-250 sample additive (Thermo Scientific, #BN2004) in a new PCR tube. Afterwards, 15 µL of the resulting mixture was reserved for BN-PAGE and while the remaining 2 µL was reserved for detecting APT5A levels (control protein) using SDS-PAGE. We then used an XCell SureLock Mini-Cell electrophoresis system to run the 15 µL samples on NativePAGE 3%-12% gradient Bis-Tris gels (Thermo Scientific,

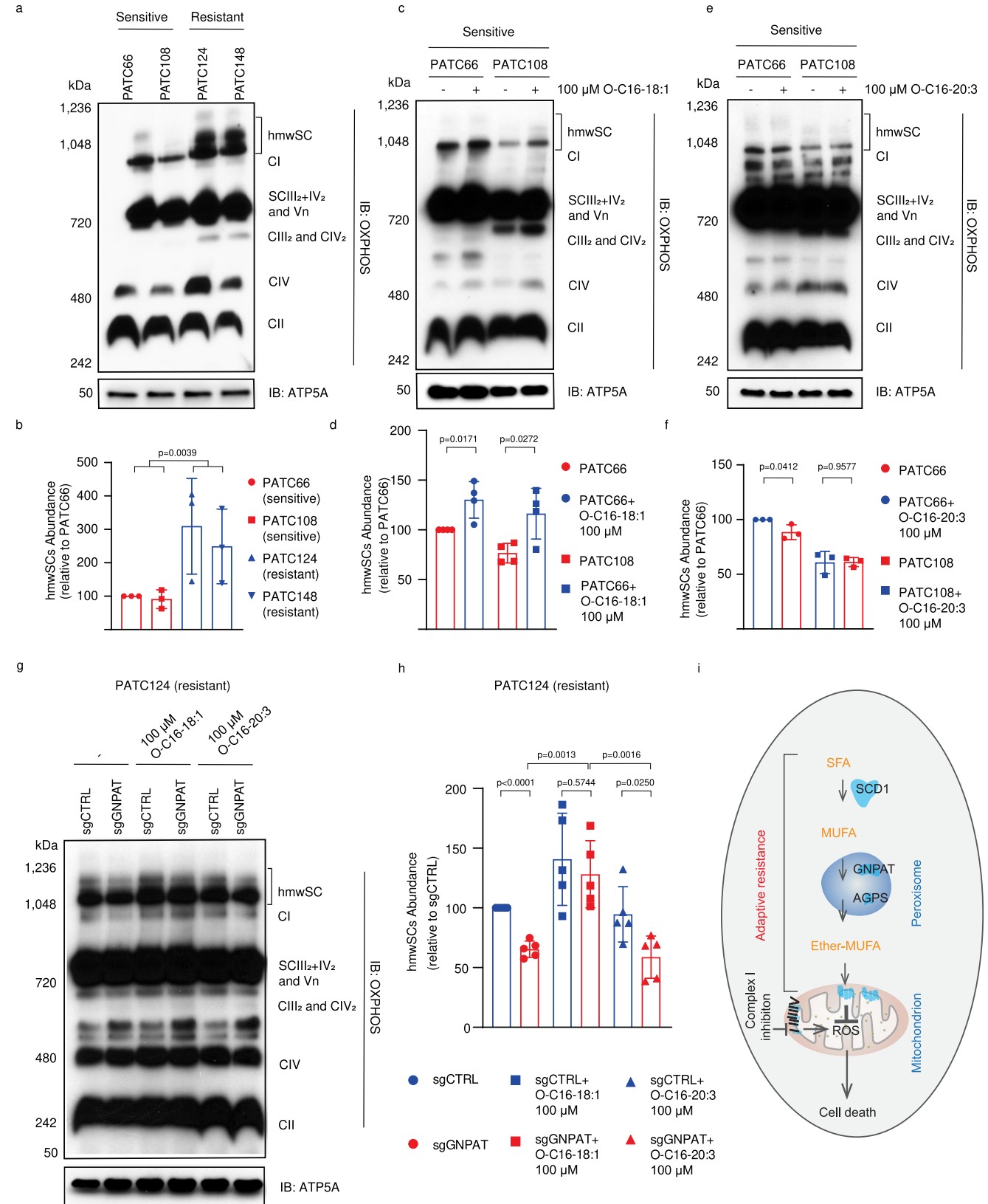

#BN1003BOX) in a 4 °C cold room. Specifically, NativePAGE anode buffer (Thermo Scientific, #BN2001) and Dark blue cathode buffer (0.2 g/L Coomassie blue G-250 in anode buffer) were used to run the gel at 150 V for 30 min. We then changed the dark blue cathode buffer to the light blue cathode buffer (0.02 g/L Coomassie blue G-250), and ran the gel at 250 V for 90 min. The iBlot gel transfer device was then prepared, and samples were transferred to a PVDF membrane with 25 V for 10 min. Subsequently, the PVDF membrane was fixed with 8% acetic acid for 5 min and then washed with $H_2O$ three times. The fixed PVDF membrane was then incubated in methanol for 5-10 min to remove Coomassie blue and then washed with $H_2O$ three times. The membrane was blocked with blocking buffer (5% milk in TBS + 0.1% Tween 20) for 30 min at room

**Fig. 5 | MUFA-linked ether phospholipids promote the assembly of mitochondrial super-complexes. a, b** BN-PAGE showed mitochondrial supercomplexes (SC) in sensitive (PATC66/108) and resistant (PATC124/148) cells (**a**). Quantifications of the high molecular weight supercomplex (hwmSC) are shown in (**b**). Data represent mean ± S.D of three biologically independent replicates. **c, d** Mitochondrial supercomplexes as shown with BN-PAGE (**c**) and quantifications of hwmSCs (**d**) in sensitive (PATC66/108) cells grown in the presence or absence of 100 μM ether-MUFA (O-C16-18:1 phosphatidylcholine (PC)) for 24 hr. Data represent mean ± S.D of 4 biologically independent replicates. **e, f** Mitochondrial supercomplexes as shown with BN-PAGE (**e**) and quantifications of hwmSCs (**f**) in sensitive (PATC66/108) cells grown in the presence or absence of 100 μM ether-PUFA (O-C16-20:3 PC) for 24 hr. Data represent mean ± S.D of three biologically independent replicates. **g, h** Mitochondrial supercomplexes as shown with BN-PAGE (**g**) and quantifications of hwmSCs (**h**) in PATC124 cells infected with control sgRNA (sgCTRL) or sgRNA targeting GNPAT(sgGNPAT) in the presence or absence of 100 μM O-C16-18:1 PC or 100 μM ether-PUFA (O-C16-20:3 PC) for 24 hr. Complex V subunit protein APT5A was detected by SDS-PAGE western blotting as control protein. Data represent mean ± S.D of 5 biologically independent replicates. **i** Schematic representation of the major conclusions. Peroxisome-derived ether phospholipids, especially those linked with MUFAs, enhance mitochondrial ETC supercomplexes assembly to maintain redox balance and promote resistance to mitochondrial complex I inhibition. Impairment of MUFAs-linked ether phospholipids synthesis by targeting SCD1, GNPAT, or APGS synergizes with mitochondrial complex I inhibition to induce mitochondrial ROS and cell death. Statistical analysis by two-tailed Students' unpaired $t$ test with significance indicated (**b, d, f, h**). Source data are provided as a Source Data file.

---

temperature and then incubated with 10 mL of the primary antibody (total OXPHOS human WB antibody cocktail) overnight with gentle shaking at 4 °C. The PVDF membrane was then washed with TBST three times and incubated with 10 mL of the secondary anti-mouse HRP-linked antibody (Cell Signaling Technologies, 1:2000, #7076 V) at room temperature for 60 min. After washing the membrane with TBST three times, membrane signaling was developed using horseradish peroxidase chemiluminescent substrate (Thermo Scientific, #34096). All western blotting data were replicated three times. Full scans of all X-films were obtained.

### In vivo study

**Xenograft models.** We used 6–8 weeks nude mice (Strain #002019, The Jackson Laboratory) for the development of subcutaneous xenograft models. All mice were housed in caging under HEPA filtered air supply and exhaust and sterile equipment conditions with a photoperiod of 6:00 a.m. to 6:00 p.m. at 72 ± 2 F and 50% ± 10% humidity. For each mouse, $3 \times 10^5$ early-passages PATCs PDX cells were suspended in 50 μL 1×HBSS (Corning, #21022CV) and mixed with 50 μL matrigel (Westnet, #356254) in a 10 mL tube on ice. The 100 μL cell suspension was then subcutaneously injected into nude mice. At 4–5 weeks after implantation, all tumors were measured with digital caliper. Mice harboring 150–200 mm³ tumors were randomly divided into groups ($n > 3$), maintained on a fasting/feeding cycle protocol (described below), and treated with indicated compounds. Specifically, mice were treated with vehicle control (0.5% methylcellulose solution), IACS-010759 (prepared in 0.5% methylcellulose solution, 5 mg/kg), or A939572 (7.5 mg/kg,) every two days when fasting via oral gavage. Body weight and tumors were measured 2 times every week with an electronic scale and a digital caliper, respectively. Tumor volumes were calculated with the formula (Length × width × width)/2. For all mice experiments, maximum tumor volumes were not exceeded 2000 mm³ to meet the standards of IACUC. All mice were operated and treated under protocols approved by M.D. Anderson's Institutional Animal Care and Use Committee (IACUC).

**Fasting/feeding cycle mice model.** We followed the fasting/feeding cycle protocols described by Elgendy et al.[18]. Briefly, mice were treated with a 24 hr fasting/feeding cycle in which they were allowed food for 24 hr (5 pm–5 pm) and then fasted for 24 hr. During each fasting cycle, indicated compounds were administered at 8 am. To verify the fasting effect, murine blood glucose was measured by a OneTouch UltraMini meter and test strips. Mice were weighed and observed carefully twice every week to ensure that mice were maintained weight and appeared healthy.

**Immunohistochemistry (IHC) analysis.** Harvested tumor samples were cut in half, immediately fixed with 4% formaldehyde for 24 hr, and then incubated with 70% ethanol. Samples were embedded in paraffin blocks (Leica ASP300S), cut into 4 μm sections, and mounted on slides to dry. Samples were deparaffinization and a sodium citrate solution was used for antigen retrieval. After boiling, endogenous peroxidase was blocked by 3% hydrogen peroxide, and non-specific sites were blocked by Rodent Block M solution (Biocare Medical, #RBM961H). Samples were incubated with indicated primary antibody diluted in incubation buffer (0.1% Triton X-100 in TBS + 3% BSA) overnight at 4 °C. Afterwards, samples were washed with TBST (0.05% Triton X-100 in TBS) three times, and incubated with a secondary antibody (ImmPRESS HRP Goat anti-Rabbit IgG Polymer Detection Kit, Peroxidase; Vector Biolabs, #MP-7451-50) for 40 min. Afterwards, ImmPRESS NovaRED substrate (Fisher Scientific, #NC9925963) was added to detect the IHC signal, and Harris Hematoxylin 100 sec was added for staining nuclei. To detect hypoxic areas using an Hypoxyprobe Omni Kit (Fisher Scientific, #NC0132724), 60 mg/kg pimonidazole HCl was administered through an intraperitoneal injection at 4 hr before tumor harvest. Following our IHC protocol, anti-pimonidazole rabbit antisera were used as the primary antibody, and Goat anti-rabbit IgG ImmPRESS secondary antibody [HRP polymer] (Vector Biolabs MP-7451-50) was incubated as the secondary antibody.

### Metabolomics and lipidomics analysis

**Metabolomics analysis.** A total of $8 \times 10^6$ PATC cells were seeded in dishes (90% confluency) and changed medium 2 hr before harvested. All media were aspirated and immediately 4 mL MeOH: H₂O 80:20 (v/v) was added to dishes. All dishes were kept at −80 °C for 10 min. Next, everything was scraped off and pipette into a 15 mL conical tube. Centrifuge the mix and aliquot volumes equivalent to the protein concentration (from parallel dishes) into 1.5 EP tubes. Keep the samples at −80 °C until shipped to Metabolon. Inc for analysis. All methods utilized followed manufacturers' instructions for using the Waters ACQUITY UPLC System for a Waters ACQUITY ultra-performance liquid chromatography (UPLC), and a Thermo Scientific Q-Exactive high resolution/accurate mass spectrometer (Thermo Scientific) interfaced with a heated electrospray ionization (HESI-II) source, and an Orbitrap mass analyzer operated at 35,000 mass resolution.

**Mitochondrial lipidomics analysis.** For the lipidomic assay, mitochondria were purified (as described above) from $5 \times 10^7$ PATC cells. For each mitochondrial pellet suspended in 100 μL ice-cold PBS solution, 5 μL of Avanti SPLASH® LIPIDOMIX® Mass Spec Standard (330707) in methanol, 3 μL of 10 mM butylated hydroxytoluene in methanol, and 242 μL of −80 °C ethanol were added. The tubes were vortexed for 5 min and then centrifuged at 17,000 $g$ at 4 °C for 10 min. Supernatants were then transferred to a Phenomenex Impact Protein Precipitation Plate (CE0-7565) and filtered through by using a vacuum manifold. The pellets obtained from centrifugation were re-extracted with 300 μL of ethanol, and the resulting supernatants were once again passed through the protein precipitation plate. Plate wells were then rinsed with 200 μL ethanol to elute residual lipids. The lipid extracts were transferred to new Simport tubes and pre-washed with methanol. 200 μL ethanol was then used to wash the collection wells of the plate. The wash solutions and extracts were combined, and samples were dried with a centrifugal vacuum concentrator. Dried samples were

reconstituted in 100 µL ethanol. Samples were analyzed for lipidomics analysis according to manufacturer's instructions for performing chromatography with a Thermo Fisher Scientific Accucore C30 column (2.6 µm, 150 × 2.1 mm).

Data analysis was performed with Thermo Scientific LipidSearch software (version 4.2.23). Statistical analyses were conducted using R (version 3.6.0). A two-factor mixed model ANOVA was used to compare biochemical levels between sensitivity, with the cell line treated as a random effect. All P values were two-tailed. P values obtained from multiple tests were adjusted using the false discovery rate (FDR). $P \leq 0.05$ was considered statistically significant, unless otherwise specified.

## $^{13}C_{18}$-Oleic acid Isotype flux tracing

PATC cells were incubated in DMEM medium containing 10% dialyzed FBS with $^{13}C_{18}$-oleic acid for 0 or 24 hr. Ethanol was added to extract ePLs from whole-cell or mitochondria samples and vortexed for 5 min, put on ice for 10 min, and then centrifuged at 18,000 g for 10 min at 4 °C. The supernatant was transferred to an autosampler vial and 10 µL was injected into an HPLC-HRMS system. Mobile phase A (MPA) was 40:60 acetonitrile:water with 0.1 % formic acid and 10 mM ammonium formate. Mobile phase B (MPB) was 90:9:1 isopropanol:acetonitrile:water with 0.1 % formic acid and 10 mM ammonium formate. The chromatographic method included an Accucore C30 column (2.6 µm, 150 × 2.1 mm) maintained at 40 °C, autosampler tray chilling at 8 °C, a mobile phase flow rate of 0.200 mL/min, and a gradient elution program as follows: 0–3 min, 30% MPB; 3–13 min, 30–43% MPB; 13.1–33 min, 50–70% MPB; 33–48 min, 70–99% MPB; 40–55 min, 99% MPB; 55.1–60 min, 30% MPB. A Thermo Orbitrap Fusion Lumos Tribrid mass spectrometer with heated electrospray ionization source was operated in positive ionization mode, with scan ranges of 600–900 m/z and a resolution of 240,000 (FWHM). A spray voltage of 3600 V was used. Vaporizer and ion transfer tube temperatures were set at 275 and 300 °C, respectively. The sheath, auxiliary, and sweep gas pressures were 35, 10, and 0 (arbitrary units), respectively. Data were analyzed using Thermo Scientific TraceFinder software (version 5.1).

## Statistical analysis

Each experiment was repeated independently at least three times unless otherwise indicated, and all data are presented as mean ± standard deviation (S.D.). Data analysis was performed using GraphPad Prism 9 and ImageJ. Two-tailed unpaired Student's t-test was used for two-group comparison and one-way ANOVA analysis was used for multiple groups comparisons. P value <0.05 was considered statistically significant.

## Reporting summary

Further information on research design is available in the Nature Portfolio Reporting Summary linked to this article.

## Data availability

There are no restrictions on data availability for all main figures and supplementary figures. Source data are provided with this paper.

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

## Acknowledgements

We thank Walter N. Hittelman at the Experimental Therapeutics department, Tomasz Zal and Anna Zal for their support with confocal microscopy at Advanced Microscopy Core, The University of Texas MD Anderson Cancer Center (UTMDACC); We thank Mikhila Mahendra and Sonal Fnu for Seahorse XF96 Analyzer protocol; Jeffrey J Kovacs for technical support on metabolites analysis; David W Dwyer and Karen C Dwyer at Advanced Cytometry & Sorting Facility at South Campus (NCI P30CA016672) for supporting flow cytometry data analysis, UTMDACC. We thank Thomas Huynh for IHC scan services at the Department of Veterinary Medicine & Surgery, UTMDACC. This project was supported by grants from NIH/NCI SPORE in gastrointestinal cancer grant P50CA221707 to A.V., H.Y. and G.F.D.; National Cancer Institute (NCI) grant R01CA214793 and NCI P01 grant P01CA117969 to H.Y.; Sewell Family Chair in Genomic Medicine to G.F.D.; the V Foundation (V2020-018) and NIH/NCI R01CA258917 to A.V.; MD Anderson Cancer Center Moon Shots Disruptive Science Grant and PanCAN Translational Grant (19-65-YAO) to W.Y.; CPRIT Training Award (RP210028) to E.Y.Y.; NCI (R01CA248160, R01CA244931) to C.A.L.; NIH grants S10OD012304-01 and P30CA016672 to Metabolomics Core Facility.

## Author contributions

Z.C., G.F.D., and H.Y. contributed to project design, data integration, and manuscript writing; Z.C., I.L.H., M.S., E.Y.Y., J.T., and L.Y. performed in vitro experiments; Z.C., S.J., Q.E.C., N.F., J.P.G., and J.R.M. contributed to in vivo study; C.A.L., J.L.R., S.S., Q.W., J.W., P.L.L., L.J.V., B.W., J.N.W., and W.Y. were responsible for metabolomics and lipidomics experiments and data analysis; A.K.D., S.G., G.G., A.V., and W.Y. contributed to manuscript editing.

## Competing interests

C.A.L. has received consulting fees from Astellas Pharmaceuticals and Odyssey Therapeutics, and is an inventor on patents pertaining to Kras regulated metabolic pathways, redox control pathways in cancer, and targeting the GOT1-pathway as a therapeutic approach. The remaining authors declare no competing interests.

## Additional information

[1]Department of Genomic Medicine, The University of Texas MD Anderson Cancer Center, Houston, TX, USA. [2]Department of Molecular and Cellular Oncology, The University of Texas MD Anderson Cancer Center, Houston, TX, USA. [3]Translational Research to AdvanCe Therapeutics and Innovation in ONcology (TRACTION), The University of Texas MD Anderson Cancer Center, Houston, TX, USA. [4]The Oncology Research for Biologics and Immunotherapy Translation (ORBIT), The University of Texas MD Anderson Cancer Center, Houston, TX, USA. [5]Institute for Applied Cancer Science, The University of Texas MD Anderson Cancer Center, Houston, TX, USA. [6]Department of Bioinformatics and Computational Biology, The University of Texas MD Anderson Cancer Center, Houston, TX, USA. [7]Department of Systems Biology, The University of Texas MD Anderson Cancer Center, Houston, TX, USA. [8]Department of Genitourinary Medical Oncology, The University of Texas MD Anderson Cancer Center, Houston, TX, USA. [9]Department of Translational Molecular Pathology, The University of Texas MD Anderson Cancer Center, University of Texas, Houston, TX, USA. [10]Department of Molecular and Integrative Physiology, University of Michigan, Ann Arbor, MI 48109, USA. [11]University of Michigan Rogel Cancer Center, University of Michigan, Ann Arbor, MI 48109, USA. [12]Department of Internal Medicine, Division of Gastroenterology and Hepatology, University of Michigan, Ann Arbor, MI 48109, USA.
✉e-mail: ZChen13@mdanderson.org; GDraetta@mdanderson.org; hying@mdanderson.org

