## [Peer Review File · Nature Communications]

Ether phospholipids are required for mitochondrial reactive oxygen species homeostasisREVIEWER COMMENTS

Reviewer #1 (Remarks to the Author); expert in lipid metabolism:

Chen et al. describe a role for ether phospholipid (ePL) biosynthesis in the protection of pancreatic cancer cells against complex I (CI) inhibition and induction of ROS. The authors propose that pancreatic cells with high levels of ePL, especially MUFA containing ones, are better protected against complex I inhibitors due to an increased amount of supercomplexes in the mitochondria. The authors use a number of complementary and robust approaches such as PDX derived cell lines with different sensitivities, protein depleting approaches (sgRNA), and small molecule and inhibitors, metabolomics/lipidomics to address the mechanism for their observations. The ability of oleate and ePL to reverse some of the effects were very interesting. Overall, the study is well executed. I have some questions and comments that I think the authors should consider.

1. One overall comment is that PDAC tumors are known to be considerably hypoxic and the control of MUFA metabolism (e.g. SCD and oleate) is strongly affected by hypoxia in pancreatic cancer cells. It would be pertinent to assess the sensitivity of the cell lines under mild hypoxic conditions (~1%) to CI inhibitors in vitro. In addition, some of the viability studies in Figure 3 on the effects of the knockouts on CI inhibitor sensitivity could also be repeated in hypoxia. It may be that the hypoxia-induced changes in MUFA metabolism could impact the sensitivity to CI inhibitors.
2. In the methods it was not clear how fluorescence intensity in Figure 2c-f was quantified? Do the replicates represent individual experiments or cells? If cells, how were they averaged and could differences in transfection efficiency influence the results (e.g. there do not seem to be any positive controls).
3. I agree with the authors that the effect of the ePL seems more closely related to mitochondria function than susceptibility to peroxidation. Most of the results with lipid peroxidation sensors were modest, but the mitochondrial ROS measurements were much more robust and reproducible throughout the studies. However, I believe that Figure 5 would benefit from slightly more clarification and elaboration. The authors used a "total OXPHOS human WB antibody cocktail" which is useful, but the blots are challenging at times to discern exactly which bands are considered supercomplexes. Some of the bands are not present in all blots, and again there seemed to be no positive controls for identification/verification of the supercomplexes themselves. The authors may want to consider re-probing for an individual protein from within the supercomplexes for cleaner, perhaps easier to interpret, western blots. For instance, an antibody to a specific subunit of complex I and/or complex III as a complementary approach to the pan-complex antibody results.
4. The authors suggest that the presence of supercomplexes promotes resistance to CI inhibition. Besides Native-PAGE blots on the supercomplex formation, are there ways to directly assess supercomplex activity, such as in-gel assays? Some type of "functional readout" would strongly support their conclusions. For instance, differences in resistant vs sensitive cell lines or that the function of these supercomplexes is less sensitive to interference by CI inhibitors?
5. The use of arachidonic acid as a PUFA supplement in Ext Fig. 6 is ok, but maybe not ideal given that AA is a precursor for a number of bioactive lipids (e.g. eicosanoids, prostanoids, leukotrienes, HETEs, etc) which could indirectly influence the results of the experiments. Perhaps consider using mead acid as an alternative?
6. One key topic that does not appear to be addressed is if there is enhanced biosynthesis of ePL in resistant cell lines? What is the underlying reason for this difference? Is there higher expression of ePL biosynthetic enzymes? Are there more peroxisomes? PPAR activity? The authors should attempt to address this in some manner.
7. Perhaps I overlooked this, but did the authors ever show that exogenous supplementation with ePL led to their enrichment specifically in the mitochondria?

Minor comments:

Few typos throughout

Authors suggest that ePL impact supercomplex activity is novel, but others have also investigated this area using similar approaches (see Bennet CF et al. Nat Chem Biol 2021 and associated references contained therein).

Were the glucose-limited conditions actually glucose-deprived? Was the medium changed during the assay? Did the cells consume all of the 5.5mM glucose after 4 days (Fig. 1 and Ext Figs. 1 and 2)?

Reviewer #2 (Remarks to the Author); expert in pancreatic cancer and metabolism:

The work by Chen et al studies the contribution of ether phospholipids to mitochondrial function in pancreatic cancer PDXs. Authors classify pancreatic cancer PDXs into sensitive/resistant with respect to their sensitivity to mitochondrial complex I inhibition. They focus on the inhibitor IACS-010759, whose effects were studied in vitro and in vivo. In fact, authors determine that mitochondrial ROS production after treatment is key for the sensitivity to the compound, and this parameter is dependent on monosaturated fatty acids. Specifically, authors determine that synthesis of ether phospholipids in the peroxisome supports the assembly of mitochondrial supercomplexes in resistant cells.

This is an interesting work in the field of pancreatic cancer metabolism and therapeutic targeting. However, there are some key points needing further clarification.

Major points

1. My main concern involves the very basis of this work: the definition of sensitive/resistant PDXs in terms of cell death considering the variability of the results shown. For instance, the PDX PATC53 (considered sensitive) shows around 70% cell death in Fig. 1A, 50% in Fig.4J and 40% in Fig. S9B. This percentage matches with the cell death detected for the resistant PDX PATC124 in Fig. 3. Indeed, a lot of variability can be observed comparing the different figures for the same PDX; however, the results within the same panel for the same PDX are very consistent with very low standard deviations. How is this possible? While the two groups are very well defined for metformin response or even phenformin, the results for IACS-010759 are less clear (Fig.S1).
2. In my opinion, including side-by-side comparison of sensitive vs resistant cells in terms of basal and mitochondrial OCR, intracellular ROS levels and, most importantly, ETC efficiency, is necessary to support the foundations of this work and strengthen the conclusions. Indeed, resistant cells should show increased ETC efficiency and diminished ROS production. The same parameters should be studied in sensitive cells after ether-MUFA/PUFA incubation (Figure 5)
3. Mitochondrial ROS measurements should be expressed as median of the population, not as percentage of positive cells: all the cells are necessarily producing mitochondrial ROS at some degree. This is particularly important for example in figure 2b, where data showing the change in the median ROS levels after treatment with respect to basal would be necessary to evaluate the effects of the compound.
4. It is necessary to include cell death/ROS kinetics after IACS-010759 treatment since, for example, experiments shown in figure 1c, d, e, f and figure 5 are after 24h of treatment. At that time point, cell death is likely very low but changes in ROS should be detectable.

Minor points

1. Including labels Sensitive/Resistant cells along the figures would help interpretation of the results.
2. It would be helpful to include additional fields of the pictures shown in Fig. 1L
3. Please, include a table compiling the main features (i.e. mutations) of the PDXs used in the study
4. Please, indicate the IC50 for IACS-010759 for each cell line used in the paper

Reviewer #3 (Remarks to the Author); expert in mitochondrial metabolism and oxidative stress:

In this manuscript, Chen et al. identify a role for MUFA-linked ether phospholipids in mitochondrial ROS homeostasis and cellular response/sensitivity to the complex I inhibitor IACS-010759. They further correlate reduced abundance of mitochondrial respiratory supercomplexes with sensitivity to IACS-010759, which can be reversed with MUFA- but not PUFA-linked ether phospholipids.

Overall, the combination of chemical and genetic approaches to manipulate ether phospholipids, along with rescue experiments with different ether phospholipid species provide strong support for the authors' conclusions. The PDAC PDX models and PDX-derived cell lines with differential sensitivity to IACS-010759 also make for a powerful experimental system. However, given the known role of ether lipids in management of oxidative stress/lipid peroxidation and the relevance of peroxisome-derived ether lipids in supercomplex assembly, the advancements provided by the current manuscript is somewhat limited. In particular, the data in Fig. 5 are underdeveloped and not well integrated with the rest of the paper (detailed below). Additionally, a deeper dive into mechanisms of cell death in this system will strengthen the paper.

Specific comments:

1. The data in Fig. 5 are interesting but correlative and do not address whether differences in supercomplex assembly/abundance is the mechanism underlying sensitivity/response of cells to IACS-010759. Are strategies to boost SC assembly (independent of MUFA-linked ether phospholipids) sufficient to render cells resistant to IACS-010759? What is the authors' explanation as to how MUFA-linked ether lipids stabilize SCs?
2. Is the protein abundance of SC components and assembly factors different across IACS-010759 sensitive and resistant lines?
3. Overall, it will be helpful to provide a better characterization of the PDAC lines in terms of their OXPHOS dependency. Do these lines show differences in survival when cultured in galactose media? Or under hypoxia? Are there differences in mitochondrial mass? It would also be helpful to discuss/mention K-RAS and other mutational status of the PDAC cells and if/how this tracks with resistance/sensitivity profiles.
4. Difference in response to IACS-010759 is only observed in low glucose conditions, which the authors interpret as a compensatory role for glucose when complex I is inhibited. Does IACS-010759 still increase mitochondrial ROS in sensitive lines cultured in glucose-rich medium?
5. A deeper dissection of cell death mechanisms (ferroptosis etc.) in the context of IACS-010759 and how it is countered by MUFA-linked ether lipids will strengthen the current study. Here, the authors could also take advantage of biopsies from their xenografts to assess appropriate markers beyond caspase 3.
6. Do MUFA-linked ether lipids similarly rescue death induced by other complex I inhibitors?

Other Comments:

7. Information on the concentration of different lipid species used in for rescue experiments appears to be missing.
8. The paper will benefit from a better description of statistical methods used for the different comparisons, including number of independent repeats/experiments.

RESPONSE TO REVIEWERS' COMMENTS

The revised sections of the manuscript and responses to reviewer comments are marked in blue.

Reviewer #1 (Remarks to the Author); expert in lipid metabolism:

Chen et al. describe a role for ether phospholipid (ePL) biosynthesis in the protection of pancreatic cancer cells against complex I (CI) inhibition and induction of ROS. The authors propose that pancreatic cells with high levels of ePL, especially MUFA containing ones, are better protected against complex I inhibitors due to an increased amount of supercomplexes in the mitochondria. The authors use a number of complementary and robust approaches such as PDX derived cell lines with different sensitivities, protein depleting approaches (sgRNA), and small molecule and inhibitors, metabolomics/lipidomics to address the mechanism for their observations. The ability of oleate and ePL to reverse some of the effects were very interesting. Overall, the study is well executed. I have some questions and comments that I think the authors should consider.

1. One overall comment is that PDAC tumors are known to be considerably hypoxic and the control of MUFA metabolism (e.g. SCD and oleate) is strongly affected by hypoxia in pancreatic cancer cells. It would be pertinent to assess the sensitivity of the cell lines under mild hypoxic conditions (~1%) to CI inhibitors in vitro. In addition, some of the viability studies in Figure 3 on the effects of the knockouts on CI inhibitor sensitivity could also be repeated in hypoxia. It may be that the hypoxia-induced changes in MUFA metabolism could impact the sensitivity to CI inhibitors.

We thank the Reviewer for this insightful suggestion. To evaluate the impact of hypoxia, we treated our patient-derived xenograft (PDX) lines with 10 nM IACS-010759 under mild hypoxic conditions (1% O₂) for 30 hours. Similar to our observations under normoxia, complex I inhibition induced more cell death in sensitive PDX lines (PATC66/108) compared to that in resistant PDX lines (PATC124/148) (RFig. 1a). In the revised manuscript these data are included in the new Supplementary Fig. 3b and are described under the subheading “Complex I inhibition identifies a latent vulnerability in a subset of PDAC models” within the Results section.

We also evaluated the impact of hypoxia on the role of ether phospholipid synthesis. Similar to our data obtained under normoxia, *GNPAT* knockout also promoted cell death in the resistant line, PATC124, following IACS-010759 treatment under hypoxia (**RFig. 1b**). Our data thus indicate that a hypoxic environment does not significantly alter the sensitivity of PDAC cells to complex I inhibitors.

RFigure 1

RFigure 1. Cell viability assay assessing response of sensitive (PATC66/108) and resistant (PATC124/148) PDX lines (a) or PATC124 cells infected with control sgRNA (sgCTRL) or sgRNA targeting *GNPAT* (sgGNPAT) (b) to IACS-010759 (10 nM) for 30 hours under hypoxic conditions (1% O₂). Data represent mean ± S.D. N=3. Statistical analysis used Student's unpaired t test with significance indicated (****p < 0.0001).

2. In the methods it was not clear **how fluorescence intensity in Figure 2c-f was quantified**? Do the replicates represent individual experiments or cells? If cells, how were they averaged and could differences in transfection efficiency influence the results (e.g. there do not seem to be any positive controls).

We have described quantification of fluorescence intensity in the Materials and Methods under the subheading, "Induction of mitochondrial oxidative stress determines sensitivity to complex I inhibition," PATC66 cells were independently transfected with two H₂O₂ probes for 24 hr, pCS2+MLS-HyPer7 that localizes in mitochondria or pCS2+HyPer7-NES that localizes in the

cytoplasm, followed by DMSO or IACS-010759 treatments. After fixation and staining with mitochondrial (TOMM20) and nuclei (Hoechst) markers, we used a wide-field Nikon Eclipse-Ni microscope to perform image analysis on 10-20 different areas from 3 biological replicates in each group. We then used MATLAB (The MathWorks, Inc) to quantify the H₂O₂ probe signal (GFP). We set up a threshold value to cut off auto-fluorescence interference and measured the GFP-positive or Hoechst-positive regions in each image. GFP intensity in dot plots was normalized by dividing the size of each GFP-positive area by the size of the Hoechst-positive area, such that each dot represented the relative GFP/Hoechst ratio per field. For each treatment, at least 50 GFP-positive cells from 3 biological replicates were calculated (**Fig. 2c-f**).

Two independent positive controls were included by treating pCS2+MLS-HyPer7 groups with mito-PQ (mitochondria-targeted redox cycler) and pCS2+HyPer7-NES groups with Paraquat (PQ, cellular ROS inducer). As expected, mitoPQ treatment selectively elevated ROS levels in mitochondria and PQ treatment promotes ROS production in the cytoplasm (**Supplementary Fig. 8**).

3. I agree with the authors that the effect of the ePL seems more closely related to mitochondria function than susceptibility to peroxidation. Most of the results with lipid peroxidation sensors were modest, but the mitochondrial ROS measurements were much more robust and reproducible throughout the studies. However, I believe that Figure 5 would benefit from slightly more clarification and elaboration. The authors used a “total OXPHOS human WB antibody cocktail” which is useful, but **the blots are challenging at times to discern exactly which bands are considered supercomplexes**. Some of the bands are not present in all blots, and again there seemed to be no positive controls for identification/verification of the supercomplexes themselves. **The authors may want to consider re-probing for an individual protein from within the supercomplexes for cleaner, perhaps easier to interpret, western blots. For instance, an antibody to a specific subunit of complex I and/or complex III as a complementary approach to the pan-complex antibody results.**

To respond to the Reviewer’s suggestions, we conducted Western blot analysis using antibodies for individual electron transport chain (ETC) complex subunits, including NDUFB8 (C-I), SDHB (C-II), UQCRC2 (C-III), COXII (C-IV), and ATP5A (C-V), and confirmed the specificity of these bands to their corresponding bands obtained from the OXPHOS antibody cocktail (**RFig. 2**). These

data are also included in the new **Supplementary Fig. 20a**, and are described under the subheading “MUFA-linked ether phospholipids promote assembly of mitochondrial supercomplexes” in the Results section of the revised manuscript. Please see also Point 4 below for a further functional validation.

RFigure 2. Mitochondrial high molecular weight supercomplexes (hmwSCs) were extracted in PATC124 cells and six equal samples were run with blue native PAGE in the same gel. The sample was then transferred to one PVDF membrane and divided into six parts, each of which were incubated with either an anti-OXPHOS antibody cocktail, anti-NDUFB8 antibody (complex I targeting), anti-SDHB antibody (Complex II targeting), anti-UQCRC2 antibody (complex III targeting), anti-COXII antibody (Complex IV targeting), or anti-ATP5A antibody (Complex V targeting).

4. The authors suggest that the presence of supercomplexes promotes resistance to CI inhibition. Besides Native-PAGE blots on the supercomplex formation, **are there ways to directly assess supercomplex activity, such as in-gel assays?** Some type of “functional readout” would strongly support their conclusions. For instance, differences in resistant vs sensitive cell lines or that the function of these supercomplexes is less sensitive to interference by CI inhibitors?

To address this suggestion, the activity of mitochondrial hmwSCs was further probed using in-gel assays (suitable for complex I, II, and IV components), as reported by Jha P, et al. (Curr Protoc

Mouse Biol. 2016. PMID: 26928661). Among the electron transport chain (ETC) complexes, complex I is robustly enriched in hmwSCs components and exhibits high activity in NADH substrate-based in-gel assay. Higher supercomplexes activity was detected in resistant lines (PATC124/148) than in sensitive lines (PATC 66/108) (RFig. 3), which is consistent with blue native blot results. These data are included in the new **Supplementary Fig. 20b**, and described under the subheading “MUFA-linked ether phospholipids promote assembly of mitochondrial supercomplexes” within the Results section in the revised manuscript.

RFigure 3. BN-PAGE shows mitochondrial hmwSCs activity and mass in sensitive (PATC66/108) and resistant (PATC124/148) cells. One hundred micrograms of mitochondria were extracted from sensitive lines (PATC66/108) and resistant lines (PATC124/148) separately, ran the gel through BN-PAGE, and then incubated with complex I substrate NADH (0.1 mg/ml) for 30 min in room temperature. 10% acetic acid was used to stop all reactions. hmwSCs containing complex I are indicated in BN-PAGE as purple bands. 50 μ g mitochondria were then extracted from the BN-

PAGE gel to measure the mass of supercomplexes by incubating with anti-OXPHOS and anti-ATP5A (control) antibodies.

5. The use of arachidonic acid as a PUFA supplement in Ext Fig. 6 is ok, but maybe not ideal given that AA is a precursor for a number of bioactive lipids (e.g. eicosanoids, prostanoids, leukotrienes, HETEs, etc) which could indirectly influence the results of the experiments. Perhaps consider using mead acid as an alternative?

To follow up on the Reviewer's suggestion, we treated the resistant cell line, PATC124, with 100 μ M mead acid (FA20:3) in the presence or absence of 10 nM IACS-010759 for 2 days. We found that mead acid treatment significantly enhanced complex I inhibition-induced cell death in PATC124 cells (R**Fig. 4**). These data are included in the new **Supplementary Fig. 11h** and described under the subheading "MUFAs are critical for ROS homeostasis and survival following complex I inhibition" in the Results section of the revised manuscript.

RFigure 4. PATC124 cells were treated with DMSO or 10 nM IACS-010759 in the presence or absence of 100 μ M mead acid for 2 days. Propidium iodide (PI) staining was used to detect cell

death events by flow cytometry. Data represent mean \pm S.D. N=3. Statistical analysis by Student's unpaired t test with significance indicated (***, $p < 0.0001$).

6. One key topic that does not appear to be addressed is if there is enhanced biosynthesis of ePL in resistant cell lines? What is the underlying reason for this difference? Is there higher expression of ePL biosynthetic enzymes? Are there more peroxisomes? PPAR activity? The authors should attempt to address this in some manner.

We thank the reviewer for the insightful comments. While we did not observe significant differences in the expression of ePL biosynthetic enzymes (e.g. GNPAT, AGPS, FAR1) (**Rfig. 5a**) nor peroxisome components (e.g. peroxisomal membrane protein 70 (PMP70), catalase) between sensitive and resistant lines (**Rfig. 5a**), synthesis of MUFA-linked ePL was found to be enhanced in resistant cells relative to that in sensitive cells by using $^{13}\text{C}_{18}$ -Oleic acid-mediated tracing (**RFig. 5b-c**). However, the flux of oleic acid into ePL was largely abolished upon deletion of *GNPAT* (**RFig. 5d-e**). These data are now included in the new **Supplementary Fig. 16**, and described under the subheading “Mitochondrial MUFA-linked ether phospholipids modulate sensitivity to complex I inhibition” in the Results section of the revised manuscript.

In addition, the expression of PPAR-delta, rather than -alpha or -gamma, is consistently elevated in resistant lines compared to sensitive ones (**Rfig. 5f**). We are currently investigating the role of PPAR-delta for the induction of MUFA-linked ePL synthesis and delineating its potential downstream targets. If referee agrees, findings from these experiments will be described in a future manuscript, as they are beyond the scope of the current study.

RFigure 5

RFigure 5. (a) Western blotting shows ether lipids biosynthetic enzymes (GNPAT, AGPS, FAR1) and peroxisome components (PMP70, catalase) compared between sensitive (PATC66/108) and resistant (PATC124/148) cell lines. (b-e) PATC124/108 (b-c) and PATC124 cells infected with control sgRNA (sgCTRL) or sgRNA targeting *GNPAT* (sgGNPAT) (d-e) were incubated with

labeled oleic acid with ^{13}C (**Oleic acid- $^{13}\text{C}_{18}$**) for indicated times. The labeled/unlabeled ePLs (PC 16:0 O/18:1[$^{13}\text{C}_{18}$], PC 16:0 O/18:1[$^{12}\text{C}_{18}$]) were detected and normalized by harvested cell number. (f) PPAR-alpha, -delta, -gamma transcriptional level detected by real-time PCR (b) between sensitive and resistant lines. Data represent mean \pm S.D. N=3. Statistical analysis by Student's unpaired t test with significance indicated (****p < 0.0001).

7. Perhaps I overlooked this, but **did the authors ever show that exogenous supplementation with ePL led to their enrichment specifically in the mitochondria?** We agree this is an important experiment, as it has been shown that exogenous ether lipids preferentially accumulate in mitochondria (Kuerschner L, et al, 2012, PLoS One; PMID 22348073). Our findings confirm that supplementation with exogenous ePL leads to a significant increase in ether phosphatidylcholine, but not total phosphatidylcholine, in purified mitochondria (**RFig. 6**). These data are included in the new **Supplementary Fig. 15**, and described under the subheading “Mitochondrial MUFA-linked ether phospholipids modulate sensitivity to complex I inhibition” in the Results section of the revised manuscript.

RFigure 6. PATC66 cells were incubated with or without 100 μM exogenous ePL (O-C16-18:1 PC) for 12 hr. Mitochondria from PATC66 were extracted and assessed for enrichment of total

phosphatidylcholine (PC) (16:0-18:1) and ether-PC (O-16:0-18:1). Data were normalized by extracted mitochondrial mass. Data represent mean \pm S.D. N=3. Statistical analysis by Student's unpaired t test with significance indicated (****p < 0.0001).

Minor comments:

8. Few typos throughout

Thank you. We have corrected all typos and indicated corrections with blue text.

9. Authors suggest that ePL impact supercomplex activity is novel, but others have also investigated this area using similar approaches (see Bennet CF et al. Nat Chem Biol 2021 and associated references contained therein).

We thank the Reviewer for pointing out this interesting study. While Bennet et al showed that peroxisome-derived ether lipids promote respirasome assembly, the contribution of the conjugated fatty acyl chain to this mechanism was not investigated. Here, our data suggest that the saturation state of the fatty acyl chain plays a major role in determining the impact of ePL on super-complex assembly. Bennet et al is now cited in our revised manuscript.

10. Were the glucose-limited conditions actually glucose-deprived? Was the medium changed during the assay? Did the cells consume all of the 5.5mM glucose after 4 days (Fig. 1 and Ext Figs. 1 and 2)?

The medium was not replaced and glucose in the medium was exhausted after 3 days following DMSO treatment and after 2 days upon complex I inhibition. Importantly, the level of glucose deprivation was comparable between sensitive and resistant lines (**RFig. 7**), indicating that the differential response to complex I inhibition was not due to differences in glucose deprivation levels. These data are included in **Supplementary Fig. 2b**, and are described under the subheading "Complex I inhibition identifies a latent vulnerability in a subset of PDAC models" in the Results section of the revised manuscript.

RFigure 7

RFigure 7. Glucose concentration in the medium was measured in sensitive lines (PATC66/108) and resistant lines (PATC124/148) treated with DMSO or 10 nM IACS-010759 on the indicated days. Data represent mean \pm S.D.

Reviewer #2 (Remarks to the Author); expert in pancreatic cancer and metabolism:

The work by Chen et al studies the contribution of ether phospholipids to mitochondrial function in pancreatic cancer PDXs. Authors classify pancreatic cancer PDXs into sensitive/resistant with respect to their sensitivity to mitochondrial complex I inhibition. They focus on the inhibitor IACS-010759, whose effects were studied in vitro and in vivo. In fact, authors determine that mitochondrial ROS production after treatment is key for the sensitivity to the compound, and this parameter is dependent on monosaturated fatty acids. Specifically, authors determine that synthesis of ether phospholipids in the peroxisome supports the assembly of mitochondrial supercomplexes in resistant cells. This is an interesting work in the field of pancreatic cancer metabolism and therapeutic targeting. However, there are some key points needing further clarification.

Major points

1. My main concern involves the very basis of this work: **the definition of sensitive/resistant PDXs in terms of cell death considering the variability of the results shown**. For instance, the PDX PATC53 (considered sensitive) shows around 70% cell death in Fig. 1A, 50% in Fig.4J and 40% in Fig. S9B. This percentage matches with the cell death detected for the resistant PDX PATC124 in Fig. 3. Indeed, a lot of variability can be observed comparing the different figures for the same PDX; however, the results within the same panel for the same PDX are very consistent with very low standard deviations. How is this possible? While the two groups are very well defined for metformin response or even phenformin, the results for IACS-010759 are less clear (Fig.S1).

We thank the Reviewer for raising these concerns. The variation in cell death levels among different experiments was due to the different duration of treatments. For example, in PATC53 cells, which are classified as sensitive to IACS-010759 treatment, we observe 68.5% cell death after 3 days (**Fig. 1A**), 52.3% cell death after 2.5 days (**Fig. 4J**), and 36.1% cell death after 2 days (**Supplementary Fig.9B**) of treatment with 10 nM IACS-010759. In PATC124 cells, which are classified as resistant to IACS-010759 treatment, we observe 31.1 % cell death after 3 days (**Fig.1a**) and 22.4% cell death after 2 days (**Fig. 2g**) of treatment with 10 nM IACS-010759 as summarized below in **RFig. 8**.

RFigure 8. Comparison of cell death in PATC53 (sensitive) and PATC124 (resistant) cells in response to 10 nM IACS-010759 treated over indicated durations (2/2.5/3 days). N=3. Data represent mean \pm S.D.

2. In my opinion, including side-by-side comparison of sensitive vs resistant cells in terms of basal and mitochondrial OCR, intracellular ROS levels and, most importantly, ETC efficiency, is necessary to support the foundations of this work and strengthen the conclusions. Indeed, resistant cells should show increased ETC efficiency and diminished ROS production. The same parameters should be studied in sensitive cells after ether-MUFA/PUFA incubation (Figure 5)

We have addressed this concern in the revised manuscript. The PDX lines are quite heterogeneous in terms of basal oxygen consumption rates, mitochondrial ROS, and ATP levels and these differences are independent from the sensitivity to complex I inhibitor in sensitive and resistant groups (**RFig. 9**). The addition of MUFA-linked ether lipid decreased basal mitochondrial ROS levels in some PDX lines (RFig, 9b), and did not increase basal ATP levels in all PDX lines (**RFig, 9c**). PUFA-linked ether lipid addition decreased basal OCR in sensitive PATC66 cells and increased mitochondrial ROS in sensitive PATC 108 cells (**RFig. 9a-b**).

These data have been added to **Supplementary Fig. 6a-c**, and are described under the subheading “Induction of mitochondrial oxidative stress determines sensitivity to complex I inhibition” in the Results section of the revised manuscript.

RFigure 9

RFigure 9. Comparison of basal oxygen consumption (a) or mitochondrial reactive oxygen species (ROS) by mito-SOX staining (b) in PATC66 / PATC108 / PATC124 / PATC148 cells with or without 200 μM O-C16-18:1 ether-MUFA or 200 μM O-C16-20:3 ether-PUFA for 1 day. c) Basal ATP levels in PATC66 / PATC108 / PATC124 / PATC148 cells with or without 200 μM O-C16-18:1 ether-MUFA for 1 day. N=3-4. Statistical analysis by Student's unpaired t test with significance indicated (*p < 0.05; **p < 0.01; n.s. = not significant). Data represent mean ± S.D.

3. Mitochondrial ROS measurements should be expressed as median of the population, not as percentage of positive cells: all the cells are necessarily producing mitochondrial ROS at some degree. This is particularly important for example in figure 2b, where data showing the change in the median ROS levels after treatment with respect to basal would be necessary to evaluate the effects of the compound.

We agree with this comment and, as suggested, we updated all presentations of mitochondrial ROS measurements as “Mean fluorescence intensity (MFI).” These changes are shown in **Fig. 2b, i; Fig. 3b, g; Fig. 4a, d, h, i; Supplementary Fig. 6b, 7a, 11g.**

4. It is necessary to include cell death/ROS kinetics after IACS-010759 treatment since, for example, experiments shown in figure 1c, d, e, f and figure 5 are after 24h of treatment. At that time point, cell death is likely very low but changes in ROS should be detectable.

To address these concerns, we measured mitochondrial ROS and cell viability in sensitive cell lines (PATC66 and PATC108) at the relatively early time point of 24 hr of IACS-010759 treatment. Our findings show that mitochondrial ROS levels, but not cell death, are enhanced by 24 hrs of treatment in these sensitive models (**RFig. 10**). These findings thus suggest that ROS levels increase prior to the emergence of cell death in sensitive cell lines.

The data are now included in **Supplementary Fig. 7** and are described under the subheading “Induction of mitochondrial oxidative stress determines sensitivity to complex I inhibition” within the Results section in the revised manuscript.

RFigure 10

RFigure 10. Sensitive cell lines, PATC66 and PATC108, were treated with DMSO or 10 nM IACS-010759 for 24 hr. Mitochondrial ROS (a) and cell viability (b) were detected by mito-ROS and propidium iodide staining, respectively. N=3. Statistical analysis by Student’s unpaired t test with significance indicated (***) $p < 0.001$; n.s. = not significant). Data represent mean \pm S.D.

Minor points

1. Including labels Sensitive/Resistant cells along the figures would help interpretation of the results.

We have labeled “Sensitive/Resistant” in all figures following the Reviewer’s suggestion.

2. It would be helpful to include additional fields of the pictures shown in Fig. 1L

Additional figures are included in **Fig. 1L** and **Supplementary fig. 5**.

3. Please, include a table compiling the main features (i.e. mutations) of the PDXs used in the study.

The mutational status of major oncogene/tumor suppressors of PDAC in our PDX models are now included in the new **Table 2**.

4. Please, indicate the IC₅₀ for IACS-010759 for each cell line used in the paper

The IC₅₀ (nM) of our PDXs are shown as **RTable 2**. These data are now included in the new **Table 1**.

RTable 1

PATC	PATC53 (Sensitive)	PATC66 (Sensitive)	PATC108 (Sensitive)	PATC118 (Resistant)	PATC124 (Resistant)	PATC148 (Resistant)
IC50(nmol/L)	6.7	5.1	7.1	13.8	13.5	20.2

Reviewer #3 (Remarks to the Author); expert in mitochondrial metabolism and oxidative stress:

In this manuscript, Chen et al. identify a role for MUFA-linked ether phospholipids in mitochondrial ROS homeostasis and cellular response/sensitivity to the complex I inhibitor IACS-010759. They further correlate reduced abundance of mitochondrial respiratory supercomplexes

with sensitivity to IACS-010759, which can be reversed with MUFA- but not PUFA-linked ether phospholipids. Overall, the combination of chemical and genetic approaches to manipulate ether phospholipids, along with rescue experiments with different ether phospholipid species provide strong support for the authors' conclusions. The PDAC PDX models and PDX-derived cell lines with differential sensitivity to IACS-010759 also make for a powerful experimental system. However, given the known role of ether lipids in management of oxidative stress/lipid peroxidation and the relevance of peroxisome-derived ether lipids in supercomplex assembly, the advancements provided by the current manuscript is somewhat limited. In particular, the data in Fig. 5 are underdeveloped and not well integrated with the rest of the paper (detailed below). Additionally, a deeper dive into mechanisms of cell death in this system will strengthen the paper.

We thank the Reviewer for these constructive suggestions and concerns. Our study discovered a previously unrecognized role for ether lipids and the saturation status of fatty acyl chains linked to the ether lipids in the adaptive responses of human pancreatic cancer cells to survive mitochondrial complex I inhibition. We further demonstrated that ether lipids play an important role in the regulation of mitochondrial ROS generation by controlling the assembly of hmwSCs. While our data are consistent with the recent reporting on the role of ether lipids in the formation of hmwSCs, our study contributes two novel findings: 1) the ability of ether lipids to promote hmwSC assembly is determined by the saturation status of the linked fatty acyl chain, and 2) ether lipid-mediated hmwSC assembly ~~plays a critical role~~ is a major factor in controlling mitochondrial ROS generation in response to complex I inhibitors. Also, to address the Reviewer's other point, we conducted additional experiments, detailed below, to determine the cell death type(s) induced by complex I inhibition. We hope the reviewer will consider our new data to be sufficient to address the concerns.

Specific comments:

1. The data in Fig. 5 are interesting but correlative and do not address whether differences in supercomplex assembly/abundance is the mechanism underlying sensitivity/response of cells to IACS-010759. Are strategies to boost SC assembly (independent of MUFA-linked ether

phospholipids) sufficient to render cells resistant to IACS-010759? What is the authors' explanation as to how MUFA-linked ether lipids stabilize SCs?

We thank the Reviewer for these insightful questions. CRISPR/Cas9-mediated deletion of *UQCC3*, which is involved in high molecular weight supercomplex (hmwSC) assembly, decreases the quantity of mitochondrial hmwSCs in the resistant PDX cells as well as significantly sensitizes the cells to IACS-010759 (RFig. 11). This finding indicates that hmwSC formation is important for resistance to complex I inhibition. These data are now included in **Supplementary Fig. 22a-b**, and are described under the subheading "MUFA-linked ether phospholipids promote assembly of mitochondrial supercomplexes" within the Results section in the revised manuscript.

Regarding the question of how MUFA-linked ether lipids stabilize hmwSCs, we agree with the reviewer that this is an outstanding issue that remains to be addressed. Although, we are testing several hypotheses, including the impact of ether lipids on mitochondrial membrane biology, since results are still preliminary at this stage it is too early to make any conclusive claims.

RFigure 11. a) hmwSC samples were extracted and components were detected in control (sgCTRL) and *UQCC3*-deleted (sgUQCC3#1 and #2) PATC124 cells. b) Cell viability assay

assessing control (sgCTRL) and *UQCC3*-deleted (sgUQCC3#1 and #2) PATC124 cells treated with DMSO or 10 nM IACS-010759 for 3 days. Cell viability was detected by propidium iodide staining. N=3. Statistical analysis by Student's unpaired t test with significance indicated (****p < 0.0001). Data represent mean ± S.D.

2. Is the protein abundance of SC components and assembly factors different across IACS-010759 sensitive and resistant lines?

We examined this and found that differences in protein abundance of hmwSC components were observed for complex IV (COX II), which was upregulated in resistant cells, but not for the other hmwSC components (**RFig. 12**). However, oxygen consumption rates were found to be similar between resistant and sensitive cell lines (**RFig. 9**). Furthermore, the expression levels of hmwSC assembly factors, including *UQCC3*, *HIGD1A*, *STOML2*, or *COX7A2L*, were observed to be similar between sensitive and resistant lines (**RFig. 12**). Overall, our findings suggest that basal levels of hmwSC components and assembly are similar between sensitive and resistant PATC lines.

RFigure 12

RFigure 12. Western blotting shows mitochondrial hmwSC components and assembly factors in sensitive (PATC66/108) and resistant (PATC124/148) cells.

3. Overall, it will be helpful to provide a better characterization of the PDAC lines in terms of their OXPHOS dependency. Do these lines show differences in survival when cultured in galactose media? Or under hypoxia? Are there differences in mitochondrial mass? It would also be helpful to discuss/mention K-RAS and other mutational status of the PDAC cells and if/how this track with resistance/sensitivity profiles.

To begin addressing these concerns, we first cultured sensitive and resistant PDX lines in 5.5 mM galactose with or without 10 nM IACS-010759 treatment. Our findings showed that cell viability was similar between sensitive and resistant lines when cultured with 5.5 mM galactose alone, but significantly higher cell death was observed in sensitive lines relative to resistant lines when treated with IACS-010759 (RFig. 13a). Sensitive and resistant lines were then growth under hypoxia (1% O₂) with or without 10 nM IACS-010759 treatment. Our findings reveal cell viability

is similar between sensitive and resistant lines, but as expected, significantly higher cell death was observed in sensitive lines relative to that in resistant lines when treated with IACS-010759 (**RFig. 1a-b**). Additionally, we did not observe significant differences in mitochondria mass, as measured by the protein levels of TIMM23 and TOMM20, among PDX lines (**RFig. 13b**). Finally, we did not observe correlations between sensitivity to complex I inhibition and the mutational status of key PDAC genes in our PDX models (**new Table 2**). These data have been added to the new **Supplementary Fig. 3a, 6d** and described under the subheadings “Complex I inhibition identifies a latent vulnerability in a subset of PDAC models” and “Induction of mitochondrial oxidative stress determines sensitivity to complex I inhibition” within the Results section in the revised manuscript.

RFigure 13

RFigure 13. a) PATC108 (sensitive) and PATC148 (resistant) cells were treated with DMSO or 10 nM IACS-010759 in 5 mM galactose medium for 1 day. Cell viability was detected by propidium iodide staining. N=3. Statistical analysis by Student’s unpaired t test with significance indicated (***) $p < 0.001$; n.s. = not significant). Data represent mean \pm S.D. b) Western blotting analysis shows mitochondrial mass (TIMM23, mitochondrial inner membrane protein; TOMM20, mitochondrial outer membrane protein) in sensitive (PATC66/108) and resistant (PATC124/148) cells.

4. Difference in response to IACS-010759 is only observed in low glucose conditions, which the authors interpret as a compensatory role for glucose when complex I is inhibited. Does IACS-010759 still increase mitochondrial ROS in sensitive lines cultured in glucose-rich medium?

Our findings show that complex I inhibition does not induce mitochondrial ROS or cell death under high glucose conditions (25mM) (RFigure 14). It's likely that the cells grown in such super-physiological concentration of glucose can maintain sufficient glycolysis to sustain viability upon OXPHOS inhibition.

RFigure 14. Sensitive (PATC108) and resistant (PATC148) lines were treated with DMSO or 10 nM IACS-010759 in high glucose (highglc; 25 mM) or low glucose (lowglc; 5.5 mM) medium for 3 days. Mitochondrial ROS (mito-sox) and cell viability (propidium iodide staining) were detected by flow cytometry assay. N=3. Statistical analysis by Student's unpaired t test with significance indicated (***) $p < 0.001$; n.s. = not significant). Data represent mean \pm S.D.

5. A deeper dissection of cell death mechanisms (ferroptosis etc.) in the context of IACS-010759 and how it is countered by MUFA-linked ether lipids will strengthen the current study. Here, the authors could also take advantage of biopsies from their xenografts to assess appropriate markers beyond caspase 3.

We appreciate the Reviewer's constructive suggestions. Complex I inhibition induces cleaved-caspase 3 and lipid peroxidation in sensitive lines (Supplementary Fig. 5 and Supplementary Fig. 12), thus indicating a potential role for apoptosis or ferroptosis in the cell death. To further dissect the underlying mechanism, we treated cells with various inhibitors for different types of cell death, including Z-VAD-FMK (pan-caspase inhibitor), ferrostatin-1 (ferroptosis inhibitor), liproxstatin-1 (ferroptosis inhibitor), DFO (deferoxamine, ferroptosis inhibitor), and necrostatin-1 (necrosis inhibition). Puromycin was used as the positive control for apoptosis, while erastin was used as the positive control for ferroptosis (**RFig. 15a-b**). Our findings show that treatment with Z-VAD-FMK or DFO, but not necrostatin-1, ferrostatin-1, nor liproxstatin-1, partially rescued the cell death induced by IACS-010759 (**RFig. 15c-d**), thus indicating that the cell death induced by complex I inhibition cannot be accounted for by any of the cell death mechanism alone. These data are included in **Supplementary Fig. 13**, and described under the subheadings "MUFAs are critical for ROS homeostasis and survival following complex I inhibition" with the Results section in the revised manuscript.

To demonstrate the induction of lipid peroxidation *in vivo*, immunohistochemistry (IHC) for 4HNE was conducted in xenograft tumor samples. Our findings show that complex I inhibitor treatment results in enhanced lipid peroxidation in tumors derived from the sensitive line compared to those derived from the resistant line (**RFig. 15e**). In addition, *GNPAT* deletion in xenograft tumors derived from resistant cells resulted in increased 4HNE staining following IACS-10759 treatment (**RFig. 15f**), suggesting that ether lipid biosynthesis in resistant lines offered protection against the lipid peroxidation induced by complex I inhibition. These data are included in **Supplementary figure 12b, 17b**, and described under the subheadings "MUFAs are critical for ROS homeostasis and survival following complex I inhibition" and "Mitochondrial MUFA-linked ether phospholipids modulate sensitivity to complex I inhibition" within the Results section in the revised manuscript.

RFigure 15

RFigure 15. a-c) PATC66 cells were treated with 0.5 µg/ml puromycin with or without 10 µM Z-VAD-FMK (a); 5 µM erastin with or without 200 µM DFO for 3 days (b); DMSO or 10 nM IACS-010759 with or without 10 µM Z-VAD-FMK, 200 µM DFO or 40 µM necrostatin-1 for 2.5 days (c). d) PATC108 cells were treated with DMSO or 10 nM IACS-010759, with or without 10 µM Ferrostatin-1/ Liproxstatin-1 for 3-days. Flow cytometry showed propidium iodide (PI)-positive cell death events. a-d) Statistical analysis by Student's unpaired t test with significance indicated (***) $p < 0.001$; n.s. = not significant). Data represent mean \pm S.D. e-f) 4-HNE staining of biopsies

from sensitive (PATC108) and resistant (PATC148) xenograft tumors (e) as well as from sgCTRL/sGPNPAT xenograft tumors (f) in the presence or absence of 5 mg/kg IACS-010759. Lipid peroxidation is shown. Scale bar, 200 μm .

6. Do MUFA-linked ether lipids similarly rescue death induced by other complex I inhibitors?

Our findings show that, in a sensitive cell line, treatment with exogenous MUFA-linked ether lipids is capable of rescuing cell death induced by phenformin (RFig. 16). These data are now included in Supplementary Fig. 19 and described under the subheading “Mitochondrial MUFA-linked ether phospholipids modulate sensitivity to complex I inhibition” with the Results section in the revised manuscript.

RFigure 16

RFig. 16. PATC66 (a) or PATC108 (b) cells were treated with DMSO (CTRL) or 200 μM O-C16-18:1 ether-MUFA with or without 12.5 μM phenformin for 3 days. Cell viability was measured by propidium iodide staining. N=3. Statistical analysis by Students’ unpaired t test with significance indicated (**** $p < 0.0001$). Data represent mean \pm S.D.

Other Comments:

7. Information on the concentration of different lipid species used in for rescue experiments appears to be missing.

We have added these details in the legends for Fig. 4 and 5 as well as Supplementary Fig. 15 and 16.

8. The paper will benefit from a better description of statistical methods used for the different comparisons, including number of independent repeats/experiments.

We have added additional details about the statistical analyses, including the number of biological repeats, in the Methods as well as the legends for Fig.2c-f and Supplementary Fig. 8.

REVIEWERS' COMMENTS

Reviewer #1 (Remarks to the Author):

The authors addressed my concerns. No further comments.

Reviewer #2 (Remarks to the Author):

Authors have properly addressed all the comments raised, significantly improving the manuscript.

Reviewer #3 (Remarks to the Author):

The authors have appropriately addressed my previous comments.